

# Contrasting exhumation histories and relief development within the Three Rivers Region (Southeast Tibet)

Xiong OU[1], Anne Replumaz[1], Peter van der Beek[2]

[1] Institut des Sciences de la Terre (IsTerre), Université Grenoble Alpes, Grenoble, 38058, France
[2] Institute of Geosciences, Potsdam University, Potsdam, 14476, Germany

*Correspondence to*: Xiong OU (xiong.ou@univ-grenoble-alpes.fr) and Anne Replumaz (anne.replumaz@univ-grenoble-alpes.fr)

**Abstract.** The Three Rivers Region in Southeast Tibet represents a transition between the strongly deformed zone around Eastern Himalayan Syntaxis and the less deformed southeast Tibetan plateau margin in Yunnan and Sichuan. In this study, we compile and model published thermochronologic ages for two massifs facing each other across the Mekong River in the core of the Three Rivers Region, by using the thermo-kinematic code Pecube to constrain their exhumation and relief development history. Modelling results for the low-relief, mean-elevation BaimaXueshan massif, east of the Mekong River, suggest regional rock uplift at a rate of 0.25 km/Myr since ~10 Ma, following slow exhumation at a rate of 0.01 km/Myr since at least 22 Ma. River incision accounts for only 15% of the total exhumation in the BaimaXueshan. Exhumation since ~10 Ma is significantly higher (2.5 km) than that estimated (~0.23 km) for the most emblematic low-relief or "relict" surfaces of Eastern Tibet, which are characterized by apatite (U-Th)/He ages older than the collision age (>50 Ma). We conclude that the BaimaXueshan massif, which shows younger ages (<50 Ma) that record significant rock uplift and exhumation during the Neogene, cannot be classified as a "relict surface" despite its low relief. Modelling results for the high-relief, high-elevation Kawagebo massif, to the west of the Mekong, imply a similar contribution of Mekong River incision (20%) to exhumation, but much stronger local rock uplift at a rate of 0.45 km/Myr since at least 10 Ma, accelerating to 1.86 km/Myr since 1.6 Ma. We show that the age-elevation profiles for three thermochronometers are best modeled by rock uplift on a kinked westward-dipping thrust striking roughly parallel to the Mekong River, with a steep shallow segment flattening out at depth. Thus, the strong differences in elevation and relief that characterize these massifs are linked to variable exhumation histories due to a strongly differing tectonic imprint.

## 1 Introduction

Despite its high mean elevation of ~5000 m, Tibet is mainly characterized by low relief (<1 km), with an average slope of ~5° (Fielding et al, 1994). However, topographic characteristics vary throughout the Tibetan Plateau. Central and North Tibet are characterized by extensive low-relief surfaces, with narrow, linear, and parallel mountain ranges that rise slightly above the mean elevation of the plateau, defining "positive topography" where elevation is positively correlated with relief and mean slope (Liu-Zeng et al., 2008). In those regions, structural relief that was tectonically generated during the collision





between India and Asia has been smoothed out locally by internal drainages, which transport sediments from the nearby ranges to fill local intermontane basins, but cannot evacuate sediments out of Tibet (Meyer et al., 1998). In contrast,

Southeast Tibet is characterised by "negative topography", where deeply incised valleys are separated by patches of low-relief surfaces at a mean elevation of ~4500 m, slightly lower than the average plateau height (Fig. 1a). Although only scattered remnants of these surfaces occur in the Three Rivers Region of Southeast Tibet, they gradually become more continuous northward and eventually integrate into the extensive low-relief surfaces of the plateau interior (Fig. 1b, c). The most emblematic surface with remarkable low relief occurs on the Triassic Daocheng granite, located between the Yangtze

and the Yalong rivers draining the southeastern plateau margin in Yunnan/Sichuan (Fig. 1c). These low-relief "relict surfaces" have been interpreted as remnants of a paleo-landscape that was formed at low elevation and subsequently uplifted (Clark et al., 2005; 2006). No such low-relief surfaces are observed further west, between the Salween River and the Eastern Himalayan syntaxis (Fig. 1b, c).

Samples collected from these low-relief surfaces in southeast Tibet show much older thermochronological ages than samples

from the deep river valleys that dissect them (Fig. 1b). The Daocheng plateau and other low-relief surfaces of the Yunnan/Sichuan margin are characterized by apatite (U-Th)/He (AHe) ages >50 Ma and apatite fission track (AFT) ages >100 Ma, whereas samples collected from the gorges of the Dadu, Yalong and Yangtze rivers show AHe ages <15 Ma and AFT ages <50 Ma (e.g., Clark et al., 2005; Ouimet et al., 2010). These data have been interpreted as recording a regional rapid incision phase starting between ~13 and 9 Ma in Eastern Tibet and used as a proxy for a widespread plateau uplift

shortly preceding this incision (Clark et al., 2005; Ouimet et al., 2010). However, the direct link between river incision and regional tectonic uplift that is inherent in this interpretation has been disputed (Liu-Zeng et al., 2008; 2018; Nie et al., 2018). Subsequent thermochronologic studies have provided evidence for an earlier phase of rapid exhumation, the timing of which varies regionally between 30 and 20 Ma in the Longmenshan (Wang et al., 2012; Tan et al., 2014), between 40 and 30 Ma in the Yalong thrust belt (Zhang et al., 2016), and between ~60 and ~40 Ma in BaimaXueshan massif (Liu-Zeng et al., 2018).

This earlier phase has also been linked to uplift of the Southeast Tibetan plateau, as paleo-elevation data suggest it has been close to its present-day elevation since the Late Eocene – Oligocene (Hoke et al., 2014; Li et al., 2015; Wu et al., 2018).

An inherent problem with thermochronology data is that they do not provide direct information on (either rock- or surface-) uplift (England and Molnar, 1990; Reiners, 2007). However, such data do potentially provide constraints on paleo-relief (Braun, 2002; Reiners, 2007) and careful thermo-kinematic modelling of thermochronological datasets may allow

differentiating between regional exhumation, presumably related to rock uplift, and relief change through valley incision (e.g., Valla et al., 2010; 2011). Spatial patterns of thermochronological ages also potentially allow constraining the kinematics of rock exhumation and the underlying structural geometry (e.g., Robert et al., 2011; Braun et al., 2012).

In this paper, we use the thermo-kinematic modelling code Pecube (Braun et al., 2012) to quantify both river incision and structurally controlled rock exhumation, based on a dense thermochronologic dataset from the Three Rivers Region. The

Three Rivers Region forms the transition between the Eastern Himalayan syntaxis and the southeast margin of the Tibetan





plateau in Yunnan, where the Salween (Nu), Mekong (Lancang) and Yangtze (Jinsha) rivers run closely parallel to each other over hundreds of kilometers and deeply dissect the plateau margin (Fig. 1b). The narrow spacing between the rivers has been interpreted as resulting from strong lateral shortening in response to indentation of the Indian-plate corner (Hallet and Molnar, 2001; Yang et al., 2015). In the core of this region, the Mekong River separates the Kawagebo massif (max.

elevation 6740 m) to the west from the BaimaXueshan massif (max. elevation ~5200 m) to the east. The BaimaXueshan massif corresponds to the southern prolongation of the elongated low-relief surface of Markam (mean elevation ~4500 m), located upstream between the Yangtze and the Mekong rivers (Fig. 1c). This transition region, separating areas without low-relief surfaces to the west and southwest from areas with extensive low-relief surfaces to the east and north, is key to better understand plateau-growth mechanisms and the geodynamic processes operating both within the high strain zone around the

syntaxis and within the lower strain zone of the Yunnan/Sichuan margin (Fig. 1b). Here, we quantify the exhumation history of the low-relief BaimaXueshan massif during the collision period to identify the different roles played by regional rock uplift and river incision and we compare it with the exhumation history of the high-relief Kawagebo massif, while constraining the geometry of the crustal fault responsible for uplifting the latter.





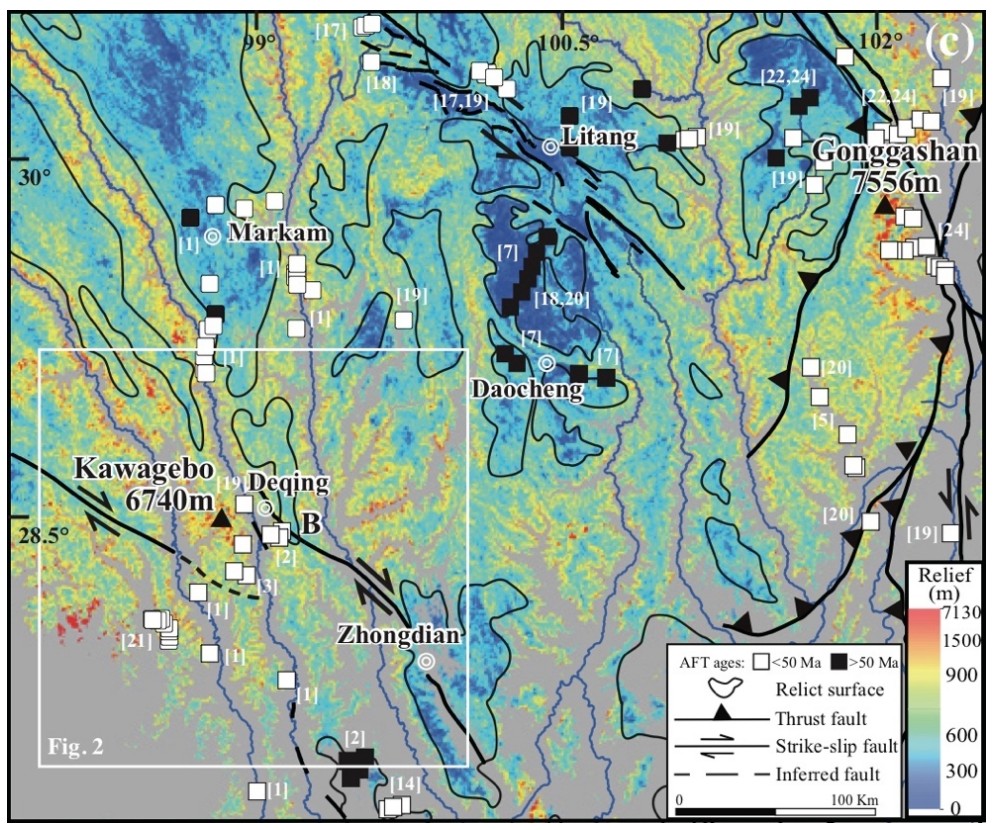


Figure 1: Tectonic and geomorphic setting of Southeast Tibet. (a) Topography and tectonic framework of the central and eastern Tibetan Plateau, based on DEM from USGS. White box shows extent of Fig. 1b; red box shows extent of Fig. 1c. (b) Main faults, rivers, relict surfaces and Apatite (U-Th)/He (AHe) data, overlain on Landsat image mosaic of Southeast Tibet (USGS). (c) Relief map (modified from Zhang et al., 2015), derived from 30-m resolution digital elevation model (DEM). 1-km radius relief is shown for pixels at elevations >3500m above sea level. Grey and black outlines delimit relict surfaces, as identified by Clark et al. (2006). Apatite (U-Th)/He (AHe) and Apatite Fission track (AFT) ages are from [1] Yang et al. (2016), [2] Liu-Zeng et al. (2018), [3] Replumaz et al. (2020), [4] Ouimet et al. (2010), [5] Clark et al. (2005), [6] Zhang et al. (2016), [7] Tian et al. (2014), [8] Nie et al. (2018), [9] Godard et al. (2009), [10] Wang et al. (2012), [11] Xiao et al. (2015), [12] Zeitler et al. (2014), [13] Gourbet et al. 2019, [14] Cao et al. (2020), [15] Shen et al. (2016), [16] Wang et al. (2018), [17] Zhang et al. (2015), [18] Reid et al. (2005), [19] Wilson and Fowler (2011), [20] Lai et al., (2007), [21] Lei et al. (2006), [22] Zhang et al. (2017), [23] Dai et al. (2013), [24] Xu and Kamp (2000). Abbreviations: B: BaimaXueshan, EHS: Eastern Himalaya Syntaxis, RRF: Red River fault, TRR: Three Rivers Region, XSHF: Xianshuihe fault.

## 2 Geologic setting

### 2.1 Cenozoic tectonic evolution of Southeast Tibet

At the regional scale, the mean elevation between the Eastern Himalayan syntaxis and the Mekong River (>5000 m) is

higher than the mean elevation of the low relief surfaces east of the river (~4500 m), suggesting higher shortening around the syntaxis (Fig. 1a,b). Within this zone, the Kawagebo massif, near the city of Deqing, peaks at 6740 m and towers over the plateau by more than 1700 m. Across the Mekong River in this area, a remarkable topographic and geomorphologic contrast is observed between the Kawagebo massif to the west, showing a high topography, high-relief and rugged landscape (Fig.





2c), and the BaimaXueshan massif to the east, characterised by a mean plateau elevation (~4500 m) and a low-relief (Fig.
2b). Furthermore, deeper and larger U-shaped glacial valleys are observed in the Kawagebo massif than in the
BaimaXueshan.

From the late Eocene to the early Miocene, the dominant driver of deformation in SE Tibet was the extrusion of the
Indochina block along the left-lateral AilaoShan ductile shear zone following the Red River (Fig. 1b), subsequently inverted
along the right-lateral Red River fault since ~5-10 Ma (Leloup et al., 1995, 2001; Replumaz et al., 2001; Fyhn and Phach,
2015). High-temperature thermochronologic data from SE Tibet shear zones show ages mainly between ~34 and ~17 Ma,
along the Red, the Salween and the Mekong rivers, which have been linked to this extrusion process (Leloup et al., 2001;
Wang et al., 2016, 2018). In contrast, no clear estimation for the age of plateau uplift in the Three Rivers Region has been
obtained. Rapid sediment filling of the Jianchuan basin, located downstream to the Three Rivers Region, around 37-35 Ma
demonstrates significant erosion in the source region just predating extrusion (Gourbet et al., 2017). This basin has
subsequently experienced significant deformation and exhumation along thrust faults between ~28 and 20 Ma (Cao et al.,
2019).

In the Three Rivers Region, the AilaoShan-Red River shear zone joins the Mekong River, following a distinctive Jurassic
red-wine colored clastic formation (Fig. 2a), which is intensively sheared. Intense shear and deformation is attested by
mostly steep to subvertical stratification and fault planes along the Mekong River, particularly in the Jurassic unit (Replumaz
et al., 2020). Mostly right-lateral sub-horizontal striations have been recognized, incompatible with either motion along the
left-lateral AilaoShan-Red River ductile shear zone or thrusting. Parallel to this shear zone along the Mekong River, several
thrusts affecting Eocene basins have been observed, but the timing and amount of exhumation related to activity on these
thrusts have not been fully quantified (Liu-Zeng et al., 2018; Replumaz et al., 2020). The roughly north-south trending
thrusts and shear zones have been left-laterally offset, with the most prominent offset occurring north of the Kawagebo
massif (Fig. 2a). Some of these faults have been reactivated as right-lateral faults, such as the Zhongdian fault crossing the
BaimaXueshan massif near Deqing. Present-day deformation of SE Tibet is dominated by active right-lateral strike-slip
faults, such as the Jiali fault, splitting eastward into the Parlung and PoQu faults (Fig. 1b), which accommodate overall
clockwise rotation around the Eastern Himalayan syntaxis (Gan et al., 2007; Bai et al., 2018). In the Three Rivers Region,
the Parlung fault jumps north-eastward to the Zhongdian fault, uplifting the Kawagebo massif in a restraining overstep since
at least ~10 Ma (Fig. 2a), then jumps south-eastward to the Red River fault, opening the Lijiang pull-apart basin (Replumaz
et al., 2020).





Figure 2: (a) Geological map of the Kawagebo area in the Three Rivers Region. Active faults are in red, other major faults in black.
Thermochronology data are shown with colour according to age and symbol according to method. K: Kawagebo massif; B: BaimaXueshan massif. (b) View of the low-relief BaimaXueshan massif at ~4500 m mean elevation, with summits peaking at ~5200 m visible at the





background. (c) View of high-relief Kawagebo massif, with summit peaking at 6740 m, while the Mekong River valley lies at ~2000 m elevation, creating almost 5000 m of relief.

## 2.2 Timing and quantification of exhumation and incision along the Mekong River

In the upper and lower reaches of the Mekong River, AHe ages from the valley bottom cluster around 20-15 Ma and have been interpreted as recording incision of the river since ~17 Ma linked to intensification of monsoonal precipitation (Nie et al., 2018). However, Oligocene or earlier (>34 Ma) entrenchment of the lower reach of the Mekong River has previously been proposed, based on river offsets by structures related to the extrusion of Indochina in Burma (Lacassin et al., 1998). Regionally, a northward increase in erosion rate along the Mekong River and a westward increase toward the syntaxis have

been proposed since ~10 Ma (Yang et al., 2016).

In the BaimaXueshan massif, located east of the river in its middle reach, numerous AHe ages between 5 and 16 Ma, AFT ages between 20 and 60 Ma, and zircon (U-Th)/He (ZHe) ages between ~80 and 120 Ma have been reported by Yang et al. (2016) and Liu-Zeng et al. (2018). Age-elevation profiles have been interpreted as recording two rapid phases of exhumation, one between 60 and 40 Ma, interpreted as being linked to the main regional uplift and crustal thickening phase,

the other since ~20 Ma, speculated to be linked to regional Miocene river incision (Liu-Zeng et al., 2018). However, quantitative time-temperature modelling of these age-elevation profiles using the 1D QTQt model (Gallagher, 2012) leads to a different scenario, with rapid exhumation only starting at 10 Ma, slowing down between 8 and 2 Ma, and accelerating since 2 Ma (Replumaz et al., 2020).

In the Kawagebo massif, also in the middle reach of the Mekong River but to the west, younger AHe ages between 1 and 4

Ma, AFT ages between 3 and 7 Ma, and a single ZHe age at ~8 Ma have been reported from the Mekong valley and its tributary Yanzhi valley (Yang et al., 2016; Replumaz et al., 2020). Quantitative time-temperature inversion of the Kawagebo ages suggests rapid exhumation since at least 8 Ma, with an acceleration since ~1.5 Ma, but with no clear estimate of the onset of this rapid exhumation phase (Replumaz et al., 2020). Rapid exhumation of the Kawagebo massif has been interpreted to be linked to tectonic uplift along a local thrust fault located along the Mekong River, in a restraining stepover

between the Parlung and Zhongdian strike-slip faults (Replumaz et al., 2020; Fig. 2). However, this local structure, inferred to be related to reactivation of regional north-south trending thrusts, has not been documented in the field. Therefore, additional work is needed in this region to resolve the exhumation history of the low-relief mean-elevation BaimaXueshan and the high-relief high-elevation Kawagebo massifs during the collision period, in order to distinguish the effects of regional plateau uplift, incision of the Mekong River, and uplift along local tectonic structures.

## 160 3 Methods: thermo-kinematic modelling of multiple thermochronology data

We synthesized published AHe, AFT and ZHe ages in the Three Rivers Region around Deqing (Table S1, Supplementary Material) to constrain our finite-element thermo-kinematic modelling using Pecube (Braun et al., 2012). This model is



designed to solve the three-dimensional heat-transport equation in a crustal block undergoing lateral and vertical rock-particle transport, exhumation, and surface change. The flexural response to surface change is also taken into account,

leading to additional rock uplift and exhumation. Default thermal and flexural parameters are shown in Table 1; some of these are optimized during the inversion. In forward mode, this set of model parameters together with a tectonic/geomorphic scenario (exhumation rates, transition times, fault geometry, surface evolution) results in rock-particle paths and an evolving 3D crustal temperature field through time, which are used to predict cooling ages for different thermochronometers. These predicted cooling ages are then compared to measured ages, defining a misfit value for this set of parameters as expressed

by:

$$\emptyset = \sqrt{\sum_{i=1}^{N} \frac{(\alpha_{i,obs} - \alpha_{i,pre})^2}{\sigma_i^2} \Big/ N} \qquad (1)$$

where N is the number of data points, $\alpha_{i,obs}$ and $\alpha_{i,pre}$ are the observed and predicted ages for data point $i$, respectively, and $\sigma_i$ is the uncertainty on the age for data point i. Inversion using the Neighbourhood Algorithm is employed to search for a best-fitting geological scenario, i.e., the set of model parameters leading to the minimum misfit value, as well as to define the

resolution with which these parameter values can be constrained (Braun et al., 2012). Modelling results are represented in scatter plots for different parameter couples and as 1D or 2D marginal probability density functions (pdf's) of each individual parameter (Braun et al., 2012).

Geological scenarios in Pecube take into account temporal variations in exhumation rates and in paleo-topography, separately or together. Here, we define a "steady-state topography scenario" as a scenario considering only spatially and

temporally varying rock exhumation rates, but sustaining the modern topography without any evolution through time. In contrast, we define an "incision scenario" as a scenario considering a simple topographic evolution from a plateau at a prescribed elevation to the modern topography, with possible different phases separated by transition times. In this paper, in order to explore the influence of incision of the Mekong River, we set up an initial plateau at 4500 m elevation, which corresponds to the average elevation of low relief surfaces in SE Tibet (e.g. Markam and Daocheng surfaces), evolving

toward the present-day topography characterized by the deeply incised Mekong River valley (Fig. 2c). A combined scenario starting with a plateau but allowing additional rock exhumation phases is defined as a "plateau scenario", where transition times mark the onset timing of river incision from the initial plateau as well as different exhumation phases. To model spatial variations of uplift above a crustal-scale fault, a 3D fault geometry is implemented following Robert et al. (2011), defining a "tectonic scenario", where the topography is in steady state, and rock exhumation is due to motion along the fault. The fault

trace at the surface and the number of segments at depth are fixed parameters in this inversion. The geometry of the fault is represented by different segments, separated by several inflection points, which are defined by their coordinates (r, s), corresponding to the depth and distance relative to the surface trace of fault, respectively. Those coordinates and velocities are inverted to constrain the fault geometry and exhumation history best fitting the thermochronology dataset.





| Thermal parameters | Value | Reference | Plate Flexural Parameters | Value |
|---|---|---|---|---|
| Crustal thickness | 50 km | Yao et al. (2010) | Crustal density | 2700 kg/m³ |
| Thermal diffusivity | 25 km²/Ma | Braun et al. (2012) | Mantle density | 3200 kg/m³ |
| Sea level temperature | 25 °C | Bermúdez et al. (2011) | Young's modulus | $1.10^{11}$ Pa |
| Atmospheric lapse rate | 4 °C/km | Bermúdez et al. (2011) | Poisson ratio | 0.25 |
| Crustal heat production | 10 °C/Ma | Braun et al. (2012) | Equivalent elastic thickness | 22.8 km |
| Model basal temperature | 900 °C | Wang et al. (2012) | | |

Table 1: Default thermal and flexural parameter settings for Pecube modelling.

## 4 Results

The strong contrasts in elevation, relief, and thermochronological ages between the BaimaXueshan and Kawagebo massifs on the two flanks of Mekong River lead us to model their exhumation histories separately. We tested "steady-state", "incision", and "plateau" scenarios for both massifs, to resolve the influence of Mekong River incision on their exhumation history (Table 2). For the Kawagebo massif, we additionally tested a "tectonic" scenario implementing a thrust fault with a surface trace along the Mekong River, as suggested in a recent study (Replumaz et al., 2020), aiming to determine its geometry and the contribution of its activity to exhumation.

### 4.1 BaimaXueshan massif

The exhumation history of the BaimaXueshan massif has been modelled using the available data for three thermochronometers (AHe, AFT, ZHe; Fig. 3a, Table S1) from Yang et al. (2016) and Liu-Zeng et al. (2018). Models were run for 110 Ma to encompass the full history recorded by these data. We initially ran steady-state topography scenarios (Table 2). A model with two exhumation phases, with protracted slow exhumation at a rate of 0.04 km/Myr from 110 to 7 Ma followed by rapid exhumation at a rate 0.42 km/Myr since 7 Ma, fits the entire dataset with a relatively low misfit (3.29). However, the break-in-slope in AFT ages around 40 Ma, marked by three ages at elevations between ~4000 and ~4700 m and interpreted by Liu-Zeng et al. (2018) as recording a rapid phase of exhumation, is not reproduced (Fig. 3a). Moreover, AHe ages >10 Ma, marking a subtle break-in-slope around 4500 m elevation, and AFT ages <25 Ma are not well modelled either in this steady-state topography scenario (Fig. 3a, 3b). To test the existence of a rapid exhumation phase around 40 Ma, as proposed by Liu-Zeng et al. (2018), we modelled more complex scenarios with three or four exhumation phases and specified transition times. The best-fitting of these models comprises a four-phase scenario (Table 2, Fig. S1), resulting in a slightly lower misfit (3.13) than the previous two-phase scenario. However, none of the more complex models resolve a rapid exhumation phase around 40 Ma; they all result in best-fit models characterized by slow exhumation that accelerated in the last few Myr, very similar to the two-phase exhumation scenario (Table 2). We conclude that the data do not require rapid exhumation prior to the last few Myr and we do not further consider these more complex scenarios.





Second, we ran an incision scenario, comprising a plateau at an elevation of 4500 m at the beginning of the model (110 Ma),
which linearly evolves toward the present-day topography after a transition time. The best-fitting model shows a transition
time at ~10 Ma (Table 2). For this end-member incision scenario, only the lowest-elevation AHe and ZHe ages are relatively
well predicted, while the predicted AHe ages at > ~3300 m elevation and the AFT ages do not fit the observed ages (Fig. 3a),
as expressed by a high overall misfit of 12.6. The high misfit and over-prediction of all but the lowest-elevation ages implies
that the total amount of exhumation is underestimated in this scenario.

We finally tested a plateau scenario, including both regional rock uplift and incision. As steady-state models adequately
predict the ZHe data with a protracted slow exhumation phase early in the history, but they do not reproduce the younger
ages as well, we concentrate here on the younger history, since 22 Ma, fitting only the AHe and AFT ages. The best-fit
model includes initial protracted slow exhumation at a rate of 0.01 km/Myr, a transition time at ~10 Ma, marking the onset
time of river incision, and more rapid exhumation at a rate of 0.25 km/Myr since ~10 Ma. This model has the lowest misfit
(1.16) and the posterior probability density functions (pdf's) of parameter values shows that the parameters are well resolved
(Fig. 4). Of course, this lower misfit cannot be directly compared to the misfit of the steady-state scenarios, which
compromise more data in the inversion, notably the ZHe ages. Still, this plateau scenario better reproduces the AHe and AFT
ages, including the slope break around ~10 Ma in the AHe age-elevation plot (Fig. 3b). The difference in rock uplift during
the second phase (2.94 km in the steady-state topography scenario versus 2.5 km in the plateau scenario), implies that
Mekong River incision accounts for ~15% of the exhumation recorded in the BaimaXueshan massif since ~10 Ma.

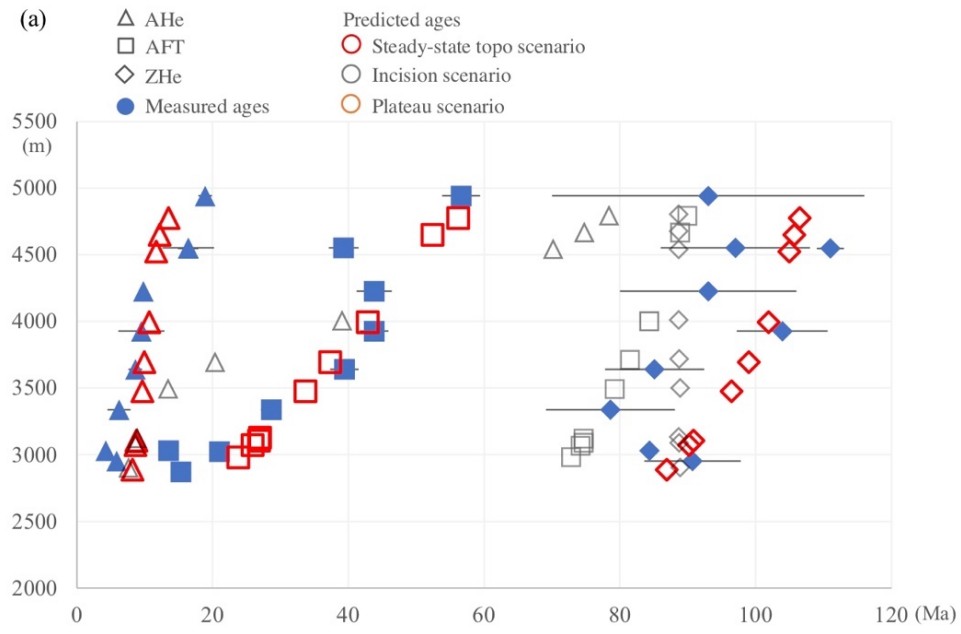



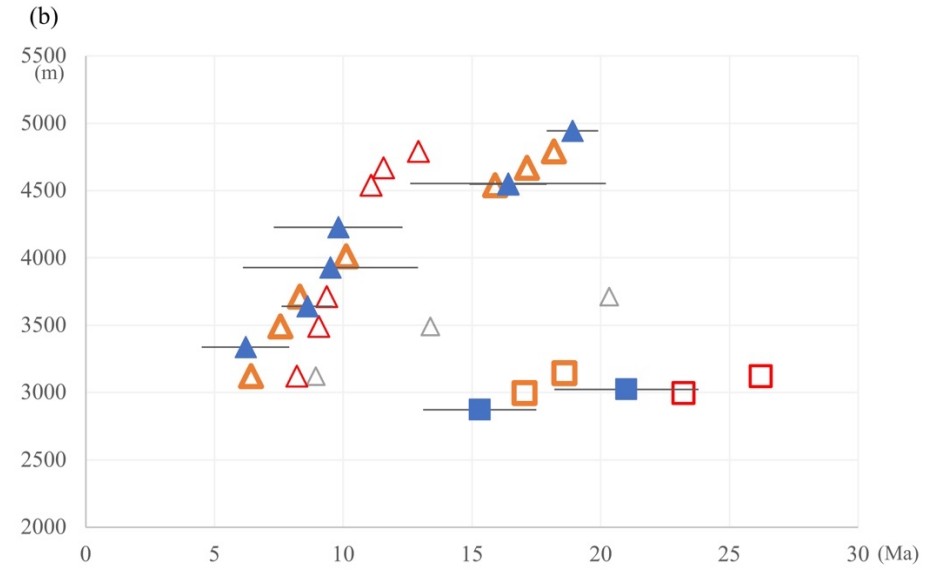

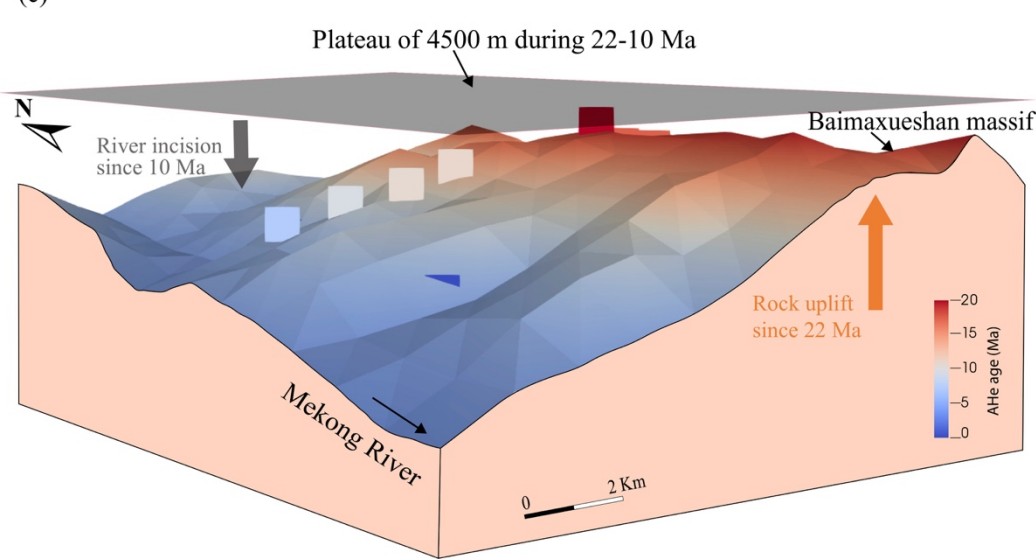

Figure 3: Model results for the BaimaXueshan massif. (a) Age-elevation profiles of measured (solid blue symbols; triangles, squares and diamonds corresponding to AHe, AFT and ZHe ages respectively) and predicted (open symbols coloured according to scenario) ages for the steady-state topography and incision scenarios. (b) Age-elevation profile of measured AHe and AFT ages (<22 Ma) and predicted ages for the plateau scenario, and predictions of the scenarios shown in (a) for comparison. (c) 3D view of the present-day topography (at resolution of 900 m) of the BaimaXueshan massif above the Mekong River, coloured according to predicted AHe ages for the plateau scenario, the best-fit model for BaimaXueshan transect (orange symbols in b). Note that the plateau is placed slightly higher than it's real elevation for convenience. The squares indicate the sample locations.





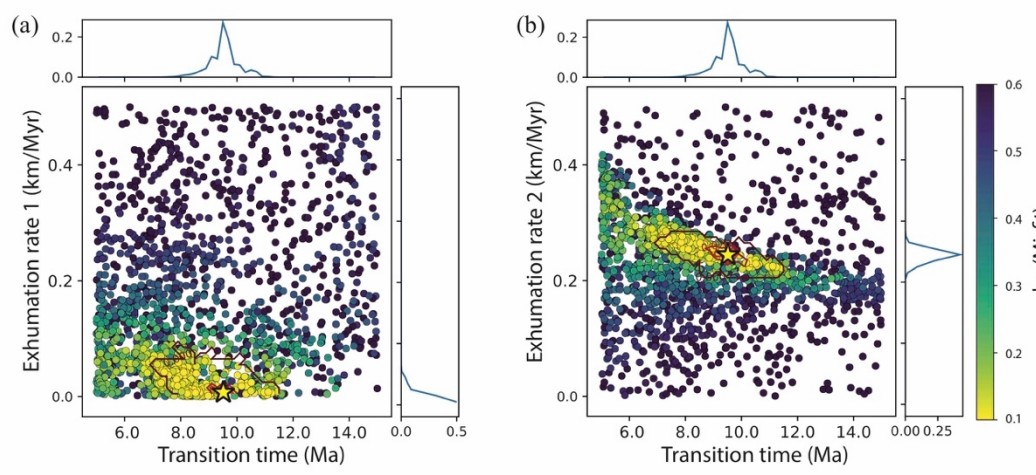

Figure 4: Scatterplots of Pecube inversion misfits for the plateau scenario model of the BaimaXueshan massif. Coloured dots represent individual forward-model runs plotted in 2-dimensional projections of the parameter space, with colours corresponding to misfit values, shown on a log scale. Posterior probability density functions for parameter values are plotted along the axes. The best-fit solution is represented by a yellow star with 2σ (dark red) and 1σ (light red) confidence contours. (a) Transition time (between first and second exhumation phase) versus exhumation rate during the first phase. (b) Transition time versus exhumation rate during the second phase.





| Massifs | Scenarios | Parameter range | Annotation | Best-fit parameter | Misfit |
|---|---|---|---|---|---|
| **BaimaXueshan** | Steady-state topography scenario (2 phases) (①: initial topography ②: modern topography B: BaimaXueshan) | E1: 0-1 km/Myr<br>T: 0-40 Ma<br>E2: 0-1 km/Myr<br>T° : 900° | Exhumation rate of 1st phase<br>Transition time<br>Exhumation rate of 2nd phase | 0.04 km/Myr<br>7 Ma<br>0.42 km/Myr | 3.29 |
| | Steady-state topography Scenario (4 phases) | E1: 0-0.5 km/Myr<br>T1: 70-40 Ma<br>E2: 0-1 km/Myr<br>T2: 40-20 Ma<br>E3: 0-0.5 km/Myr<br>T3: 20-0 Ma<br>E4: 0-1 km/Myr<br>T° : 900° | Exhumation rate of 1st phase<br>1st Transition time<br>Exhumation rate of 2nd phase<br>2nd Transition time<br>Exhumation rate of 3rd phase<br>3rd Transition time<br>Exhumation rate of 4th phase<br>Mode basal temperature | 0.04 km/Myr<br>64 Ma<br>0.04 km/Myr<br>25 Ma<br>0.06 km/Myr<br>3 Ma<br>0.72 km/Myr | 3.13 |
| | Plateau scenario (①: initial plateau ②: modern topography) | Plateau : 4500 m high<br>E1: 0-0.5 km/Myr<br>T: 0-20 Ma<br>E2: 0-1 km/Myr<br>T° : 900° | Exhumation rate of 1st phase<br>Transition time<br>Exhumation rate of 2nd phase | 0.01 km/Myr<br>10 Ma<br>0.25 km/Myr | **1.16** |
| **Kawagebo** | Tectonic scenario (①: initial topography ②: modern topography red line: fault K: Kawagebo) | r1: 22.5 km<br>s1: -50-0 km<br>r2: 0-22.5 km<br>s2: -50-0 km<br>E1: -1-0 km/Myr<br>T: 0-10 Ma<br>E2: -2-0 km/Myr<br>T° : 900° | Coordinates of fault inflection points<br><br>Fault-slip rate of 1st phase<br>Transition time<br>Fault-slip rate of 2nd phase | -16.9 km<br>8.4 km<br>-15.1 km<br>-0.69 km/Myr<br>1.5 Ma<br>-1.84 km/Myr | 0.96 |
| | Tectonic+Plateau Scenario (①: initial plateau ②: modern topography red line: fault) | Plateau : 4500 m high<br>r1: 22.5 km<br>s1: -16.9 km<br>r2: 8.4 km<br>s2: -15.1 km<br>E1: -1-0 km/Myr<br>T: 0-10 Ma<br>E2: -2-0 km/Myr<br>T° : 900° | Coordinates of fault inflection points<br><br>Fault-slip rate of 1st phase<br>Transition time<br>Fault-slip rate of 2nd phase | -0.45 km/Myr<br>1.6 Ma<br>-1.86 km/Myr | **0.66** |

Table 2: Inversion parameters for the different scenarios tested for the BaimaXueshan and Kawagebo massifs. The best-fit scenario, with the lowest misfit for each massif, is in bold. A range of exhumation rates is explored for different phases, separated by a transition time. Coordinates of the fault inflection points define the geometry of the fault plane, with ri and si (in km) corresponding to the depth and distance relative to the surface trace of fault, respectively. Note that the tectonic scenario includes a slip rate on the fault rather than a vertical exhumation velocity, and that upward slip on a thrust is indicated by negative values (see Braun et al., 2012 for more detail).



## 4.2 Kawagebo massif

Published thermochronology data from the Kawagebo massif are dispersed over more than 40 km along the Mekong River
valley (Fig. 2). To the north, several samples yield ages with large error bars, with AHe ages clustering between 2 and 5 Ma,
AFT between 5 and 30 Ma, and ZHe ages between 7 and 50 Ma (Wilson and Fowler, 2011; Liu-Zeng et al., 2018; Replumaz
et al., 2020). To the south, near the village of Yanzhi, along a west-bank tributary incising the Kawagebo massif, samples
show more coherent ages, with AHe ages between 1 and 4 Ma, AFT ages between 3 and 7 Ma, and a single ZHe age at ~8
Ma (Yang et al. 2016; Replumaz et al., 2020). We use this dataset from the Yanzhi valley to constrain thermo-kinematic
modelling of the massif evolution since 10 Ma. The steady-state topography scenario predicts a transition at 1.2 Ma between
slower exhumation (0.54 km/Myr) and rapid exhumation (1.75 km/Myr), with a low misfit (1.05). However, this scenario
does not reproduce the increasing AFT ages with elevation, nor the observed slope break in AHe ages in the age-elevation
plot (Fig. 5a). As the high-elevation samples were collected toward the core of the massif, farther from the Mekong River,
these characteristics of the age-elevation plot could potentially be explained by spatially varying exhumation rates due to a
curved fault at depth and/or incision of the Mekong River. To explore these possibilities, we include tectonic particle
transport along a thrust fault striking along the Mekong River and incision in the following inversions.

A tectonic scenario with a simple planar (one-segment) thrust converges to a similar misfit (1.08) as the steady-state
topography scenario (1.05). The best-fit model shows a fault dipping 85° to the west and a transition time at 1.3 Ma, with a
slow slip rate of 0.58 km/Myr before and a faster rate of 1.87 km/Myr after. In effect, due to the steep dip of the thrust fault,
this tectonic scenario is very similar to the steady-state topography scenario considering purely vertical motion (Fig. 5a).

A more complex tectonic scenario with a kinked (two-segment) thrust shows a slightly lower misfit (0.96) but much better
reproduces the observed age-elevation relationships (Fig. 5b). In this scenario, the best-fit fault geometry is a thrust with a
steep shallow segment (dipping at 65°) and a near-horizontal (dipping at 1.5°) deeper segment, below 15 km (Fig. 5c). The
transition time is 1.5 Ma, with a slip rate of 0.69 km/Myr before, increasing to 1.84 km/Myr after. The low slope in the AFT
age/elevation plot and the slope break in AHe ages are reproduced due to the spatial variation in exhumation rate above the
deep flat segment versus that above the steep shallow segment of the thrust (Fig. 5). Input parameters of this scenario are
relatively well constrained, as shown by the misfit scatterplots and the posterior pdf's of parameter values (Fig. 6).

For consistency with the modelling results of the BaimaXueshan massif across the Mekong River, we also ran a
"tectonic+plateau" scenario, adding the incision of a 4500 m high plateau to the tectonic scenario (two-segment thrust). This
tectonic+plateau scenario converges to a lower misfit (0.66), with a similar transition time (1.6 Ma), but smaller fault-slip
rates (0.45 km/Myr until 1.6 Ma, 1.86 km/Myr after) than the tectonic scenario with two thrust segments (Fig. 5b).
Compared to the total rock-uplift of 7.8 km since 10 Ma in the tectonic scenario, the maximum Mekong river incision (1.7
km) accounts for about 20% of the total exhumation since 10 Ma, a proportion similar to that of BaimaXueshan. The




similarity between scenarios with and without topographic change suggest that river incision plays only a relatively small

role in the exhumation history of the Kawagebo massif, characterised by strong tectonic forcing.

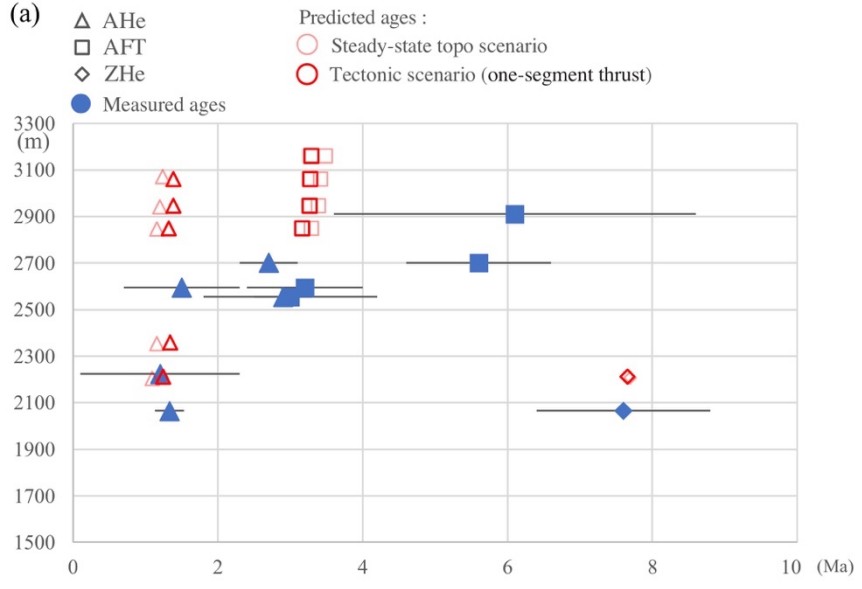

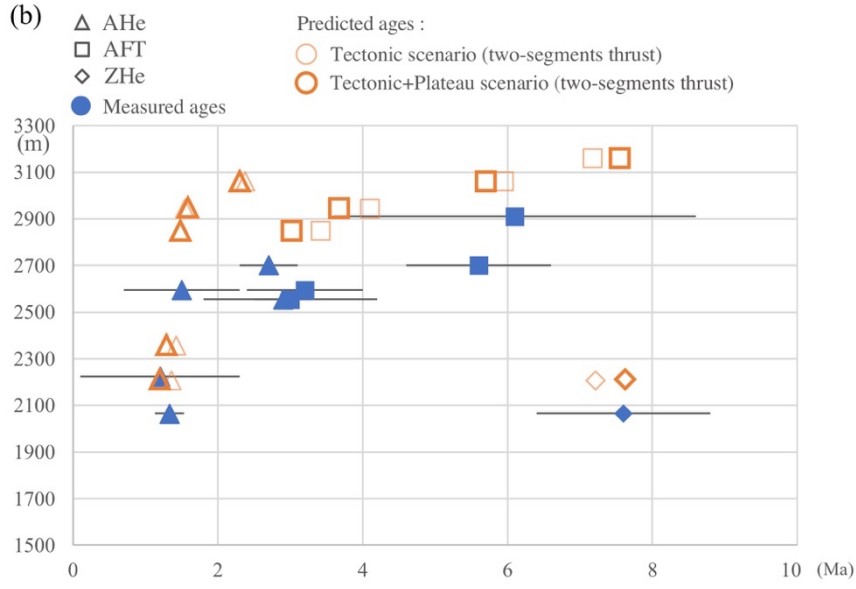



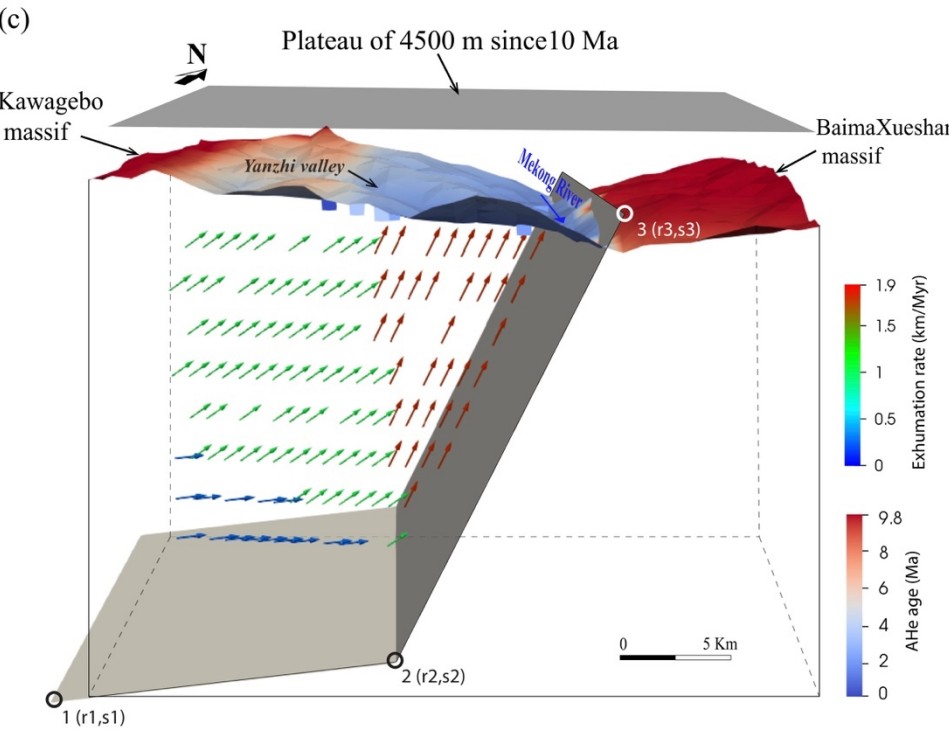

Figure 5: Modelling results for the Kawagebo massif. Age-elevation profiles of measured and predicted ages (same legend as Figure 4) for
(a) the steady-state topography scenario and the tectonic scenario with a planar (one-segment) thrust; (b) the tectonic and the
tectonic+plateau scenarios, both with a kinked (two-segment) thrust. (c) 3D view of the present-day topography of the Kawagebo massif
and the Mekong River valley, coloured according to predicted AHe ages of the tectonic+plateau scenario with a two-segment thrust. The
geometry of the thrust and rock-particle trajectories are shown at depth; squares indicate the sample locations. This scenario corresponds to
the best-fit model for the Kawagebo massif (orange symbols in b). Note that the plateau is placed slightly higher than it's real elevation for
convenience. The arrows show the velocity field generated by movement along the fault; the two-segment geometry leads to spatially
varying exhumation rates (shown by colours, scale on the right).



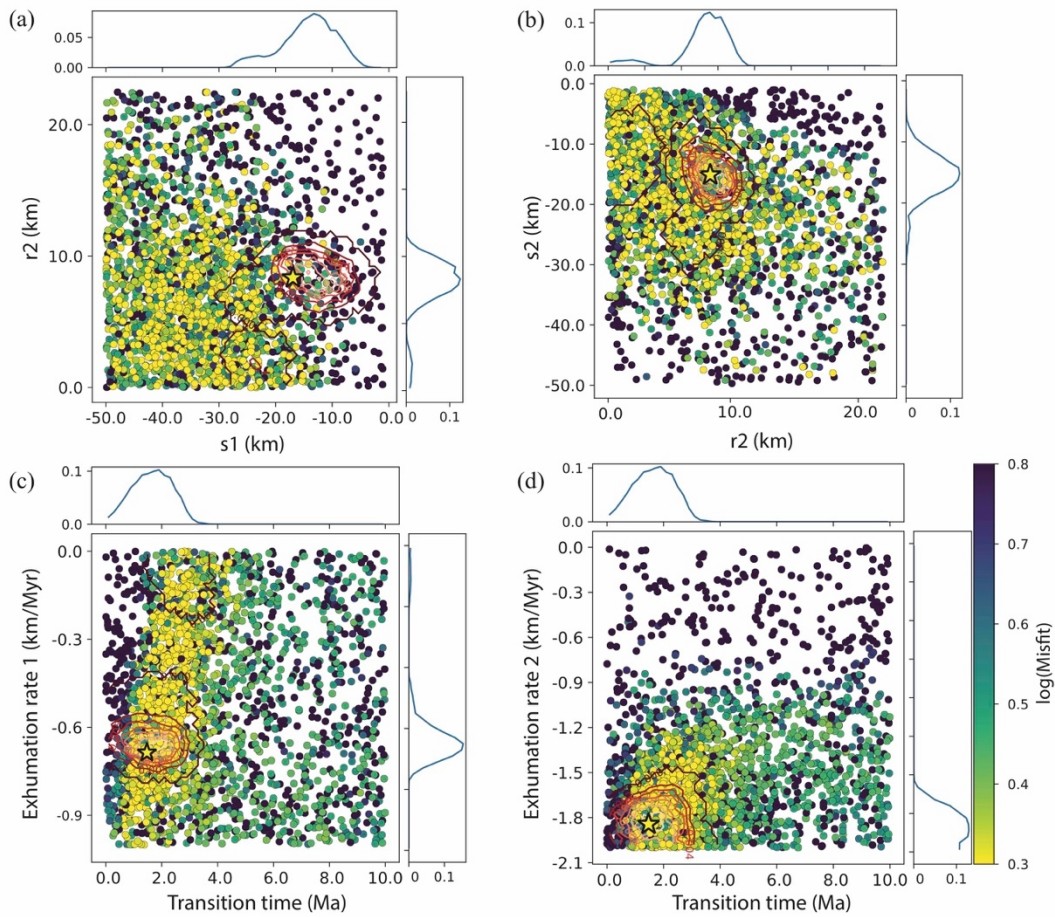

Figure 6: Scatterplots of PECUBE inversion results for two-segment thrust scenario model of the Kawagebo massif. Coloured dots represent single forward-model runs plotted in 2-dimensional projections of the parameter space, with colours corresponding to misfit values, shown on a log scale. Posterior probability density functions for parameter values are plotted along the axes. The best-fit solution is represented by a star with 2σ (dark red) and 1σ (light red) confidence contours. (a) Depth of the base of the flat (s1) versus the horizontal location of the base of the ramp, relative to the surface trace of the thrust (r2). (b) The horizontal distance of the base of the ramp (r2) versus the depth of the base of the ramp (s2). For explanation of these parameters, see Figure 5C. (c) Transition time versus slip rate along the fault during the first phase. (d) Transition time versus slip rate along the fault during the second phase.

## 5 Discussion: tectonic and climatic forcing on exhumation

### 5.1 Tectonic and climatic forcing on the evolution of the low-relief BaimaXueshan massif

For the BaimaXueshan massif, east of the Mekong River, our best-fit model shows rapid regional exhumation at a rate of 0.42 km/Myr since ~7 Ma, succeeding a phase of slow regional exhumation at a rate of 0.04 km/Myr before 7 Ma. The onset of rapid exhumation occurred slightly later compared to published thermal histories using subsets of the data and obtained using QTQt (Gallagher et al., 2012), which showed well-defined rapid exhumation at a rate of 0.26 km/Myr since 10 Ma (Replumaz et al., 2020). Our results do not support the three-phase history proposed by Liu-Zeng et al. (2018) using pseudo-



elevation profiles (Reiners and Brandon, 2006). Liu-Zeng et al. (2018) proposed rapid exhumation between 100 and 80 Ma at rate of 0.15 km/Myr, between 60 and 40 Ma at a rate of 0.6 km/Myr, and since ~20 Ma at a rate between 0.22 and 0.65 km/Myr. Our models suggest lower overall exhumation rates, do not resolve the earlier (Cretaceous and Eocene-Oligocene)

phases of rapid exhumation, and suggest that the rapid Miocene phase started since ~7 rather than ~20 Ma. Surprisingly, both the previous 1D QTQt modelling of Replumaz et al. (2020) and the 3D thermo-kinematic modelling reported here, imply that none of the tectonic events occurring in SE Tibet before 7 Ma have been recorded as exhumation phases in the BaimaXueshan massif, including extrusion of the Indochina block between ~34 and ~17 Ma (e.g. Leloup et al., 2001), significant erosion in the source region recorded by rapid sediment filling in Eocene basins during 37-35 Ma downstream of

the Three Rivers Region (Gourbet et al., 2017), or shortening of these basins between 28 and 20 Ma (Cao et al., 2019). These results suggest that tectonic forcing in this massif has been negligible in driving its exhumation history before 7 Ma.

Our best-fit model focusing on the Neogene history of the massif, since 22 Ma, implies incision of a plateau at an elevation of 4500 m since ~10 Ma, associated with regional exhumation at a rate of 0.25 km/Myr, succeeding to a phase of slow regional exhumation at a rate of 0.01 km/Myr until 10 Ma. Using such a simplified topographic evolution scenario

mimicking the incision of a plateau, we show that differential erosion between the low-relief, mean-elevation BaimaXueshan massif and the deeply incised Mekong River valley is necessary to reproduce the break-in-slope in the AHe age-elevation profile, between samples on the plateau and samples from the Mekong River valley (Fig. 3c). This scenario of incision of the Mekong River since ~10 Ma is compatible with the onset of incision estimated between 13 and 9 Ma for the Dadu and Yalong rivers gorges to the northeast (Clark et al., 2006; Ouimet et al., 2010).

Nevertheless, we also show that river incision alone is not sufficient to reproduce the AHe ages at higher elevations, as well as the AFT ages from the BaimaXueshan massif (Fig. 3a), and only accounts for ~15% of the total exhumation since ~10 Ma. For the highest samples on the plateau, our results imply that ~2.5 km of overburden was removed from the plateau surface since 10 Ma, which is an order of magnitude higher than the exhumation (~0.23 km) estimated for the low-relief Daocheng plateau to the northeast during the same timespan (Clark et al., 2005). This higher exhumation is required by the

much younger AHe and AFT ages in the BaimaXueshan massif (<18 Ma and <56 Ma respectively; Liu-Zeng et al., 2018) compared to the Daocheng massif (>50 Ma and >100 Ma respectively; Clark et al., 2005). Therefore, the BaimaXueshan massif cannot be considered to represent the same relict surface as the Daocheng granite, despite the fact that both have low relief (Fig. 1). We conclude that a relict surface sensu-stricto, i.e., defined as a surface showing little exhumation during the collision (Clark et al., 2005), should show AFT ages older than 50 Ma, and that not all low-relief surfaces in southeast Tibet

can be classified as a relict surface.

The significant removal of overburden (~2.5 km) from the BaimaXueshan massif surface (currently at a mean elevation of ~4500 m and peaking at ~5200 m) since ~10 Ma implies efficient erosion processes at high elevation have generated the currently observed low-relief landscape (Fig. 1). Glacial erosion and vigorous periglacial processes have been shown to be efficient processes that smooth high-elevation regions (Brozovic et al., 1997; Egholm et al., 2009; 2017; Hales and Roering,



2009). Zhang et al. (2016) have proposed that these processes could be active in southeastern Tibet and could lead to seemingly continuous low-relief high-elevation surfaces despite spatially differential and diachronous tectonic exhumation of 2-4 km between the surfaces. Such buzzsaw-like processes by glaciation could be active in the BaimaXueshan massif to smooth highlands, as numerous cirques, moraines and U-shaped valleys are observed across the massif, providing evidence for significant glacial erosion.

However, glacial and periglacial erosion alone cannot explain the onset of rapid exhumation at ~10 Ma, well before the onset of widespread glaciation in southeast Tibet at ~2 Ma (Fu et al., 2013). Increasing regional rock uplift toward the eastern Himalayan Syntaxis has been recorded by relatively young thermochronological ages (AHe/AFT ages < 15 Ma) in a region between the South Tibet Detachment System-Parlung fault and the Longmucuo-Shuanghu suture, which roughly follows the Mekong River (Fig. 7). This zone is also characterized by higher regional topography, with the plateau surfaces generally

exceeding 5000 m (Fig. 7). In contrast, the area to east of the Longmucuo-Shuanghu suture, including the low-relief surfaces of Markam and Daocheng, show lower average elevations and older thermochronological ages (AHe/AFT > 15 Ma). This zone between the South Tibet Detachment System-Parlung fault and the Longmucuo-Shuanghu suture is thus potentially associated with regionally enhanced Late-Miocene uplift north and east of the EHS, as argued for instance by Zeitler et al. (2014) and Schmidt et al. (2015), possibly due to the continuous northward advance of the eastern indenter corner of the

Indian plate. A north-south component of motion is also observed from the current GPS velocity field, which shows ongoing north-south shortening north of India (Fig. 7), with a vertical component of motion (Liang et al., 2013). The BaimaXueshan massif is located in the southeastern-most extremity of this regional uplift zone. Within the massif, N-S oriented thrusts like the Zigaishi thrust, east of and parallel to the Mekong River, are associated with a clear topographic crest bounding the massif to the west and could have been reactivated at a moderate rate (0.25 km/Myr) to accommodate shortening and

thickening of the Three Rivers Region since ~10 Ma (Fig. 8). Moderate uplift along those thrusts could thus be far-field effect of ongoing collision east of the EHS, which is supported by the west-to-east gradient of decreasing erosion rate across the Three Rivers Region (Yang et al., 2016).





Figure 7: (a) Simplified geological structures of SE Tibet, with GPS velocity field (modified from Wang et al., 2017), showing a N-S component of motion north of India. Blue patch is the area of average elevation above 5000 m on the Tibetan plateau. The reconstructed position of Indian continent at 10 Ma (green contour from Replumaz and Tapponnier, 2003) shows continuing convergence since that time (green north-directed arrows). (b) Correlation between thermochronological (AHe and AFT) ages in SE Tibet and topography above 5000 m (blue contour). White square represents our study area. More additional data are from Burg et al. (1998), Ding et al. (1995), Seward and Burg (2008), Tu et al. (2015), Yu et al. (2011), Zhang et al. (2015) (see Fig. 1 for references of other thermochronologic ages). Abbreviations: GF, Ganzi Fault; ITS, Indus-Tsangpo suture; JF, Jiali Fault; LMS, Longmenshan thrust system; LSS, Longmucuo-



Shuanghu Suture; MBT, Main Boundary Thrust; MCT, Main Central Thrust; PF, Parlung Fault; RRF, Red River Fault; SF, Sagaing Fault; STDS, South Tibet Detachment System; XJF, Xiaojiang Fault; XSHF, Xianshuihe Fault; ZF, Zhongdian Fault.

## 5.2 Dominant tectonic forcing on exhumation in the Kawagebo massif

Our best-fit model scenario for the Kawagebo massif combines incision of a 4500 m high plateau and tectonically controlled rock uplift above a segmented thrust (Figs. 5 and 8). Two phases have been modelled in this scenario, with an initial slip rate on the fault of 0.45 km/Myr from at least 10 Ma until 1.6 Ma, followed by rapid slip at a rate of 1.86 km/Myr since then, leading to 7.8 km of total exhumation (including 6.1 km of tectonically-driven exhumation and 1.7 km of river incision) since 10 Ma at river level (Fig. 5b). The available dataset only covers a period since 10 Ma and cannot constrain the onset timing of the initial exhumation phase. Our model outcomes corroborate earlier thermal-history modelling of the samples from the Yanzhi valley in the Kawagebo massif using QTQt, which suggested rapid cooling from 8 Ma and a significant acceleration from 1.5 Ma to present (Replumaz et al., 2020). Similar exhumation rates are predicted in a scenario that does not include river incision (transition time at 1.5 Ma, with slip rates of 0.69 km/Myr before and 1.84 km/Myr after). River incision accounts for 20% of the total exhumation of the massif (Fig. 5b). This result implies that tectonic forcing has been dominant in exhuming the Kawagebo massif since at least 10 Ma. Significant erosion rates of ~0.4-1.0 km/Myr since the late Miocene have been inferred from young AHe age of samples from near the valley bottom of the Mekong river around the EHS, showing a northward propagation of high erosion rate with time in response to the northward advance of the northeast Indian-plate indenter corner (Yang et al., 2016). Potentially, the inferred acceleration of exhumation since 1.6 Ma could be due in part to more efficient erosion related to the onset of high-altitude glaciations and/or monsoon intensification at that time. Efficient glacial erosion in the Kawagebo massif is attested to by deep U-shaped valleys, like the Yanzhi valley, suggesting >1 km of glacial erosion (Fig. 7). However, our current model only includes simplified topographic evolution scenarios and does not allow simulating such glacial incision.

Combining the 3D distribution of sample ages relative to the fault trace, fault-controlled exhumation is predicted to be linked to a steep (~65°) thrust in the upper crust, which flattens at depth and strikes roughly parallel to the Mekong River (Fig. 5c). This fault geometry reproduces the low slope in the AFT age-elevation profile as well as the observed slope break for AHe ages (Fig. 5b, c). Replumaz et al. (2020) suggested that late-Miocene uplift and exhumation of the Kawagebo massif, which peaks at ~2300 m above the mean plateau elevation of the plateau, was accommodated by such a fault. The fault was suggested to have activated at ~10 Ma in a large-scale restraining left-stepping overstep between the right-lateral Parlung and Zhongdian strike-slip faults (Fig. 7), resulting in more pronounced local uplift compared to the regional uplift recorded in the BaimaXueshan massif. Steep to subvertical stratification and fault planes are observed along the Mekong River in the sampling area (Replumaz et al., 2020), in agreement with the modelled steeply westward dipping thrust fault. However, observed striations are mostly sub-horizontal rather than dip-slip, suggesting that the active Mekong thrust is probably not outcropping in the field.





Figure 8: 3D view of the Kawagebo (K) and BaimaXueshan (B) massifs (Landsat image draped on DEM). Active faults are in red, other major faults in black. Thermochronologic ages are shown with colour representing age and symbol representing method. MK samples are from Yang et al. (2016), DQ from Liu-Zeng et al. (2018) (BaimaXueshan transect) and KW from Replumaz et al. (2020) (Kawagebo transect). Also shown are the postulated crustal geometries of faults active in the last 10 Myr. Arrows annotated B/C: positions of field photos shown in Fig. 2.

## 6 Conclusion

Using 3D thermo-kinematic models constrained by a relatively dense low-temperature thermochronology dataset, we have compared different exhumation scenarios for massifs along the middle reach of the Mekong River in the Three Rivers Region. We are able to discriminate between tectonically controlled rock uplift and river incision for both the BaimaXueshan and Kawagebo massifs. This method allows refining and validating the first-order conclusions on exhumation history inferred from age-elevation relationships and assessing the relative contributions of tectonic or climatic forcing to inferred phases of rapid exhumation.

We show that different tectonic processes dominate the exhumation histories of the two flanks of the Mekong River. To the east, the low-relief BaimaXueshan massif is not a relict surface as defined by Clark et al. (2006), as it has experienced an amount of exhumation (~2.5 km since ~10 Ma) that is an order of magnitude higher than that determined for the emblematic Daocheng relict surface (~0.23 km). None of the tectonic events affecting southeast Tibet before 10 Ma has been recorded as

exhumation phases in the BaimaXueshan massif, suggesting that tectonic forcing on the massif's exhumation has been negligible during that time. Moderate rock exhumation since ~10 Ma was mainly driven by local thrust reactivation, triggered by the northward advance of the indenting Indian plate, with incision of the Mekong River contributing only ~15% of the exhumation. The contrasting exhumation history of the Kawagebo massif to the west is dominated by strong local uplift driven by a thrust that we infer to have developed in a regional compressional overstep. Total tectonically driven

exhumation since ~10 Ma is estimated to be ~6 km, largely overprinting incision of the Mekong River too.

**Code availability**

Pecube code for thermo-kinematic modelling to invert thermochronological data is copyright of the Jean Braun. Code are available at: https://github.com/jeanbraun/Pecube

**Data availability**

The compilation of data is available in the supplementary material.

**Acknowledgements**

This work has been supported by the China Scholarship Council Funds. Jean Braun is warmly thanked for providing the Pecube code and constructive comments on the modelling.

**Author contribution**

OX, AR and PVDB have worked together through this manuscript.

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
