# Peer review of "Contrasting exhumation histories and relief development within the Three Rivers Region (Southeast Tibet)"

_Solid Earth, 2020_

## Referee Comment (RC1) · Paolo Ballato (Referee) · 4 Nov 2020

I am pleased to review the manuscript "Contrasting exhumation histories and relief development within the Three Rivers Region (SE Tibet)" by Xiong OU et al., which presents a set of 3D thermo-kinematic models to unravel the exhumation, tectonic and topographic history of two adjacent areas exhibiting a different topographic and cooling age pattern. Specifically, the Kagawebo massif is characterized by high topographic relief and very young cooling ages, while the BaimaXuexhan massif has slightly older cooling ages and is part of a system of elevated low-relief surfaces described all around SE Tibet. By testing different modeling scenarios, the authors document variations in

rock uplift rate trough time, the role of tectonic deformation, the influence of climate change and the impact of fluvial incision (the Mekong river) in controlling the observed cooling patterns.

I am not familiar with the 3D thermo-kinematic modeling, therefore, I assume that is formally correct and the selection of the modeling parameters is appropriate. Overall, the manuscript is well structured, clearly written and brings important structural and geomorphic conclusions (that probably are not properly emphasized, see comments below) and hence I believe that will be of interest for the entire scientific community. I have just few minor concerns that I think should be addressed.

1) A major assumption presented in the introduction is that the low-relief surface of the BaimaXueshan massif is part of a system of elevated low-relief surfaces observed all around SE Tibet, like that one in the Daocheng area. Is that initial assumption really correct? I think that the authors should provide a set of figures (a DEM with topographic swath profiles), to document the validity of such assumption (see also comments on line 36, Lines 366-368, and on figure 8 among others). The figures provided in this version (mostly Figure 1), seem to suggest that the low-relief landscape of the BaimaXueshan massif is different from that one of the Daocheng area. If you cannot document it, you will have to rewrite part of the introduction and the conclusions.

2) From my understanding the authors rule out the classic idea where the low-relief patches were part of the same surface that formed at low elevation and has not yet adjusted to the new rock uplift rate (see Clark and Royden), while fluvial incision should have started right after surface uplift. I think that this should be properly emphasized because it has also major geodynamic implications (for example it does not support the lower crustal flow hypothesis, or?). Following these lines: does your study add new data/constraints on the vertical and lateral growth of the Tibetan Plateau? In case, please discuss it.

3) The contribution of the incision of the Mekong river over the estimated exhumation

for both massifs should be better discussed. How did you get that estimate? In case, a swath profile across the Mekong will help to figure it out (see also comment above).

4) The erosional processes that led to the removal of ca. 2.5 km of rocks on top of the BaimaXueshan massif should be also better discussed and documented with a figure (see comments on Line 371-379). The authors mention the occurrence of landforms associated with glacial processes that I think should be documented in a figure (at least on a DEM) to strengthen their conclusions. I also think that there is an issue of time scales: rock uplift accelerated at ca. 10 Ma, while vigorous glacial erosion started only at 2 Ma. How do you reconcile this lag-time between uplift and erosion?

I hope my comment will be of help. Good job!

Line by line comments:

Lines 18-20: I think that in the abstract you should suggest what could be the cause for rock uplift and Neogene exhumation for the BaimaXuexhan massif.

Line 25. What depth (see also few comments below)?

Line 36: How do you define low-relief surfaces? You should define what kind of relief threshold you use for a low-relief surface (at least to a first order) and document it by showing a proper DEM (please see also comment on Figure 1). This probably applies also to the abstract (line 20) where the low-relief surface is mentioned for the first time.

Line 51: "However, the direct link between river incision and regional tectonic uplift that is inherent in this interpretation has been disputed (Liu-Zeng et al., 2008; 2018; Nie et al., 2018)". I think you should comment this sentence here in introduction. What has been disputed and why? Please consider, that not all readers are familiar with the debate. I guess that this is fundamental for framing your research question.

Line 61: In such regard, I think that you should already highlight in the introduction that Nie et al., interpreted the cooling pattern observed across the Mekong as an increase in precipitation rather than an indication of surface uplift (although that study is based

on samples collected further south, so the correlation may not be straightforward)

Line 100: I think that you should highlight all these differences by showing a E-W-oriented topographic swath profile across the Mekong and its flanks (BaimaXuexhan and Kawagebo). This is fundamental to properly quantify river incision (which I guess has been discussed and quantified in several papers, but readers should not be forced to look for old literature unless strictly necessary). You can provide easily such a figure. I also think that you should present a N-S topographic swath profile (see also comment to line 36 and to figure 8).

Line 107: "In contrast, no clear estimation for the age of plateau uplift in the Three Rivers Region has been obtained" In contrast to what? In the previous sentences the uplift of the plateau is not discussed. I think that a summary about our knowledge of surface uplift/paleo-elevation history of the SE Tibetan Plateau should be reported somewhere. This kind of information is completely missing in the manuscript but is fundamental for setting up your modeling scenarios.

Line 108: Please show the location of the Jianchuan basin in a figure.

Line 181: Can you model a scenario consisting of a low-relied surface at low elevation, uplifted and incised at the same time (see for example Clark and Royden 2000)? I guess that should be something like your "incision" scenario but with surface uplift., where the low-relief areas are assumed to represent a landscape that has not yet adjusted to the new rock uplift rate. Or do you think that available thermochron. data do not support any longer this idea? I think that your study may contribute to this debate.

Line 182: You assume that at 10 Ma there was already a plateau with an elevation like the modern one. Can you quickly report what data are supporting this scenario?

Line 188: What does crustal-scale mean? Please clarify it. After reading the beginning of the following section I realized that the crustal scale fault scenario was used only

for the Kawagebo Massif. I am wondering if you should anticipate this in the methods (where you talk about a generic fault) in order to provide a complete picture of the modeling scenarios (maybe the 6 lines of text at the beginning of the results section could go directly in the methods?).

Line 240: not clear to me how you quantified river incision (same applies to line 313).

Line 354-350: Long sentence. Difficult to follow. Please rewrite it

Line 360-370: I guess that this is a very important finding and tell us that the classic view (formation of low-relief surfaces at low elevation that will be uplifted and incised) decays. Should that be highlighted more? Unless the two surface are not the same object as I start suspecting (please see comment on Figure 8 and below).

Lines 366-368: "Therefore, the BaimaXueshan massif cannot be considered to represent the same relict surface as the Daocheng granite, despite the fact that both have low relief (Fig. 1)". Following my previous comments: do they really have a similar topographic relief to be considered the same object, as written in the introduction? Maybe you should also present a NE-SW topographic swath profile that cover both regions and discuss it. The relief map shown in Figure 1c does not really suggests it. I can see that the BaimaXueshan is sandwiched between 2 main rivers, so its preservation potential is lower, but your Figure 1c seems to suggest that these two features look very different.

Line 371-379: "Such buzzsaw-like processes by glaciation could be active in the BaimaXueshan massif to smooth highlands, as numerous cirques, moraines and U-shaped valleys are observed across the massif, providing evidence for significant glacial erosion". The erosional mechanisms that can remove ca. 2.5 km of rocks and generate elevated low-relief surfaces are a crucial issue. Although this is not the main point of the manuscript the conclusions of this work have important geomorphologic implications. I think that the authors should provide a figure (probably just a DEM) documenting the glacial morphologies described in the text, or at least they should refer

to previous publications documenting the processes (in Tibet or elsewhere) that may have generated such a topography. The other point that I think needs to be addressed is the timing of these erosional processes that should post-date (at least for 8 Million of years) the onset of widespread high-altitude glaciations (ca. 2 Ma). Does it mean that first you formed a much higher topography and later you removed the material without any additional, significant cooling? In case, could you document it with yours 3D thermo-kinematic models?

Line 428: "which flattens at depth" What depth? Can you provide estimates? Could they correspond to any specific rock boundary?

Figure 1B: What is the benefit of showing a Landsat image rather than a DEM? Frankly speaking I do not find such a Landasat image useful, especially, if you are not familiar with the region. The scale of figure 1a does not allow appreciating the topographic characteristics and the position of the paleosurfaces. I guess that you need to use a DEM as background of figure 1A. Especially after reading the conclusion for the BaimaXueshan massif. I also think that you need some topographic swath profiles (see comments above).

Figure 2: I am sorry, but my eyes cannot really differentiate the Triassic Qiangtang from the Yidin. Could you please increase the color difference? Also note that some black lines are thicker, apparently without any specific reason, than others. Finally, Eocene is not properly spelled; please correct it. What about showing a simplified cross section? Are there available in literature? They could help following the geological setting.

Table 2: Does the sketch has a vertical scale that I respected? What is the level of incision with respect to the initial plateau? This should be discussed somewhere (see comment to line 240) to better appreciate the contribution of the Mekong on the observed cooling history.

Figure 5: What are the modeling shortening rates over the last 1.5 Ma for the Kawagebo massif? Are these rates compatible with the lateral shearing rates recorded

by GPS?

Figure 8: From this figure the low-relief surface of the BaimaXueshan massif does not look like a typical uplifted low-relief landscape (please see also my comments above). I think that a clear definition of low-relief surface is fundamental. To me the landscape of the BaimaXueshan massif and the Daocheng area looks totally different and the large difference in cooling ages observed across these 2 surfaces is not surprising. I also suppose that the erosional mechanism that led to the development of these "surfaces" is very different (glacial vs fluvial?) and operated on different time scales. I start thinking that you are comparing surfaces that should not be compared. Such a comparison is misleading because these two surfaces are not the same landform. I understand that this is one of your major conclusions; my concern is that your starting point (i.e., these 2 surfaces are the same thing) does not appear to be supported by geomorphic evidence, unless you will document with a couple of figures.

---

## Short Comment (SC1) · 5 Nov 2020

The manuscript "Contrasting exhumation histories and relief development within the Three Rivers Region (SE Tibet)" by Xiong OU et al. provides an interesting contribute to the debate about the role of tectonics on the exhumation of Tibet. The work is based on an already existing dataset of thermochronological data that has been processed through 3D thermo-kinematic modelling. Different scenarios have been investigated, testing the relative roles of tectonics, regional uplift and localized erosion. The manuscript is well written, with a proper description of the procedures and the results and an exhaustive discussion. I have some minor issues that are detailed here below.

[Figure]

Lines 52-54. The age range for this rapid exhumation event is very large (from 20 and 60 Ma). I do not think that it is possible to talk about a single "phase" as more there one could have been taken place in a single region.

Lines 75-78. As it is written here, it is not clear if the role of tectonics in the exhumation of Kawagebo is derived from literature or is one of the output of this work. I would better specify which are the goals of the paper.

Figure 1. Separation of AFT and AHe ages in two separate maps is good in terms of readability but it forces to move from one figure to the other to have a complete picture of exhumation ages. Is it not possible to merge all the ages in a single map?

Line 85 (caption of figure 1). Why grey outlines? I see only black lines around these surfaces.

Line 103. The name "AilaoShan fault" is not in the map. Moreover, the "Red River" marks a fault and not a river.

Line 107. This sentence is not related to the previous one as they deal with very different topics. So, why "in contrast"?

Lines 108-111. This sentence is not well connected to the previous ones. In general, this paragraph appears as a collage of sentences with no clear relationships between them.

Line 112. How can a shear zone join a river? Furthermore, the AilaoShan-Red River shear zone is not marked in the map of figure 2a.

Lines 116-118. The only Eocene deposits visible in map of fig. 2A are located east of the city of Deqing. Is it just a matter of scale? In the text you describe "several thrusts affecting Eocene basins".

Figure 2A. There is a thin red line in the top of the map, nearly parallel to the Yangtze river, that is probably not correctly drawn. In the legend, check the word "Eocene". The

colors of Triassic Yidun and Qiangtang formations are very similar

Line 153. "...of the onset of this rapid exhumation phase". Are you referring to the 8 Ma or 1.5 Ma step?

Lines 155-156. Which structure? The stepover? Actually it is defined by faults and I see that these faults have been mapped and here described.

Line 158. What do you mean with "collision period"? Tectonics here was changing through time so, for example, the Eocene is marked by extensional basins.

Lines 186-187. The second part of the sentence is not very clear. Can you better explain the meaning of "transition times"?

Table 1. It is not clear if the references are related to the left or the right parameters. Or both?

Lines 222-223. This conclusion is referred to the steady-state scenarios only? Or is it more general?

Lines 230-232. I agree on the focus on AHe and AFT but... what happens if you exclude the ZHe data also in the previous scenarios? Such a change in the input data could have relevant effects on the ouputs?

Lines 289-290. Why since 10 Ma?

Line 299. Given the dipping angle, I would not use the term "thrust" for this fault.

Figure 5. I am a bit confused... Each arrow yields a color which should be related to an exhumation rate... but is this figure associated to a specific time? In fact, here you write about the presence of different exhumation phases.

Lines 345-350. I am not sure that this sentence is correct as your model is starting at 22 Ma. All the events cited here are occurring mostly before 22 Ma.

Line 352. So the paragraph above is related to the models starting before 22 Ma? This

is not very clear. But, if this is true, why are you discussing the models starting before 22 Ma if you write that these are not resolving well the dataset?

Line 368. This is not the definition of "relict surface" or, better, its definition should not be related to time of collision (Clark et al. do not give any definition like that).

Lines 385-386. Actually there are not so many data between the Parlung fault and the Longmucuo-Shuanghu suture and young ages are widespread also more to the south and to the north.

Line 388. Define the acronymn EHS.

Lines 390-391. An extrapolation of the present-day velocity field to 10 Ma ago can be chancy. On the other hand, if your results are coherent with the present-day velocity field, you can infer that plate kinematics has not changed since then.

Line 395. Not sure about the use of "far-field" as this area is along the border of EHS.

Line 413. What do you mean with "since 10 Ma at river level"?

Lines 424-426. Your model is clearly designed to verify the amount of tectonic exhumation along a thrust and the results to confirm that this model is working well. Furthermore, no acceleration of exhumation since 1.6 Ma is occurring in the BaimaXueshan massif. As a whole, these data seem to indicate that glacial erosion, if present, was of minor importance. This is an interesting outcome that could be emphasized.

Line 442. What do you mean with "postulated crustal geometries of fault"? Are you referring to the "black" faults or the active ones?

Lines 454-456. Once again I would stress the fact that data and your modelling focus on the "young" (i.e. Neogene) part of the history. So I am not very sure that you can affirm that tectonic events were negligible before 10 Ma

Line 456. Can you put a number? Otherwise the meaning of "moderate" is ambiguous.

Line 460. I would emphasize also the inferred acceleration at 1.5 Ma and the exhumation rates that jump to values higher than 1 km/Ma (one order of magnitude).

---

## Editor Comment (EC1) · Federico Rossetti (Editor) · 5 Nov 2020

The SC1 is an invited review. Accordingly, the interactive discussion for the review of the manuscript se-2020-172 can be closed with the RC1 and the SC1.

Federico Rossetti

---

## Author Comment (AC1) · 21 Dec 2020

We thank reviewer Paolo Ballato for his constructive and helpful comments. Below, we respond to the reviewer's major and line-by-line comments (response indented, line numbers in red mark the modifications in revised manuscript).

Major comments:

1) A major assumption presented in the introduction is that the low-relief surface of the BaimaXueshan massif is part of a system of elevated low-relief surfaces observed all around SE Tibet, like that one in the Daocheng area. Is that initial assumption really correct? I think that the authors should provide a set of figures (a DEM with topographic swath profiles), to document the validity of such assumption (see also comments on line 36, Lines 366-368, and on figure 8 among others). The figures provided in this version (mostly Figure 1), seem to suggest that the low-relief landscape of the BaimaXueshan massif is different from that one of the Daocheng area. If you cannot document it, you will have to rewrite part of the introduction and the conclusions.

> The assumption that all low-relief surfaces in southeastern Tibet are remnants of a paleo-landscape that was formed at low elevation and subsequently uplifted is not ours but was made in the literature previously (e.g., Clark et al., 2005; 2006). In these papers, the BaimaXueshan massif corresponds to the southern prolongation of the elongated low-relief surface of Markam (mean elevation ~4500 m). However, following our modelling results, we conclude in this paper that exhumation since ~10 Ma is significantly higher (2.5 km) in the Baimaxueshan than that estimated (~0.23 km) for the most emblematic relict Daocheng surface. We conclude that the BaimaXueshan massif, which shows younger ages (<50 Ma) that record significant rock uplift and exhumation during the Neogene, cannot be classified as a "relict surface" despite its low relief. We will try to make this more clear in our abstract and introduction.

> In Figure 1, we will change the name "relict surface" in the legend by "relict surfaces mapped by Clark", which should show a local relief <600 m. The relief is already shown on Figure 1b, but as proposed by the reviewer, we will add topographic profiles of the low-relief BaimaXueshan massif and the Daocheng granite surface (Figs. 1d, e), both showing less than 600 m local relief, but the Daocheng surface being significantly flatter than BaimaXueshan. Those profiles will also help to clarify our motivation to decipher whether they have experienced different exhumation histories. (line 85)

2) From my understanding the authors rule out the classic idea where the low-relief patches were part of the same surface that formed at low elevation and has not yet adjusted to the new rock uplift rate (see Clark and Royden), while fluvial incision should have started right after surface uplift. I think that this should be properly emphasized because it has also major geodynamic implications (for example it does not support the lower crustal flow hypothesis, or?). Following these lines: does your study add new data/constraints on the vertical and lateral growth of the Tibetan Plateau? In case, please discuss it.

> Based on the thermo-kinematic modelling showing a rapid exhumation phase since ~10 Ma, our study in Three Rivers Region does suggest that the BaimaXueshan massif was subject to Neogene tectonic uplift, which we associate with continuous indentation of the EHS. But we interpret this as relatively localised exhumation, in response to this regional uplift due to the indenting EHS, and cannot conclude on the uplift of the plateau since the onset of collision.

> We also concluded that the low relief of BaimaXueshan was related to strong erosion at high elevation, and it is not a flat paleo-landscape formed at low elevation and subsequently uplifted.

In the region near the EHS, we clearly rule out the classic idea that the low-relief patches were part of the same surface that formed at low elevation, but we conclude that only near the EHS. This study does not allow us, and it is not our goal either, to generalize this mechanism to the generation of other low-relief surfaces, such as the Daocheng surface. We therefore cannot conclude in general on the lower crustal flow hypothesis based on our study. We will rewrite part of the introduction and discussion sections of our manuscript to make this clearer.

**3) The contribution of the incision of the Mekong river over the estimated exhumation for both massifs should be better discussed. How did you get that estimate? In case, a swath profile across the Mekong will help to figure it out (see also comment above).**

Thank you for this comment, we went back and slightly modified the way we quantify river incision – this is simply the difference between the initial assumed plateau elevation at 4500 m and the present-day elevation – it therefore varies spatially, being maximum in the river valleys. We will add a schematic topographic profile across the Kawagebo and BaimaXueshan massifs to better illustrate our results (new Fig. 7). (line 250-253; Fig 7, line 405)

Note that we report the contribution of Mekong river incision to exhumation of the samples, which is different for two massifs because of different sample locations relative to the valley bottom. On the Kawagebo side, samples are from lower elevations close to the Mekong River, whereas most of the samples from Baimaxueshan are from higher elevations close to the plateau surface. But the contribution of the river incision to total exhumation in percentage is quite similar (25% versus 30% for Kawagebo and BaimaXueshan); the relative contribution is lower for Kawagebo due to the larger tectonic component of exhumation.

**4) The erosional processes that led to the removal of ca. 2.5 km of rocks on top of the BaimaXueshan massif should be also better discussed and documented with a figure (see comments on Line 371-379). The authors mention the occurrence of landforms associated with glacial processes that I think should be documented in a figure (at least on a DEM) to strengthen their conclusions. I also think that there is an issue of time scales: rock uplift accelerated at ca. 10 Ma, while vigorous glacial erosion started only at 2 Ma. How do you reconcile this lag-time between uplift and erosion?**

The erosion accompanying regional tectonic uplift from 10 Ma occurred at high elevation, slightly below the present-day glacial equilibrium line altitude (ELA, 5400 m) but above the Last Glacial Maximum (LGM) ELA (~4600 m; Fu et al., 2013). The ELA during previous glacial phases of the Quaternary would have been comparable to that of the LGM, with the average Quaternary ELA lying somewhere between these two values. Numerous cirques, moraines and U-shaped valleys are observed across the BaimaXueshan massif, providing evidence for significant glacial erosion (Fig. 8). Therefore, buzzsaw-like glacial and peri-glacial erosion, as suggested elsewhere in eastern Tibet (Zhang et al., 2016) could be a potential mechanism for in part removing this amount of overburden from the high-elevation low-relief BaimaXueshan massif, even though the extent and intensity of glaciation in SE Tibet since 10 Ma is not resolved yet. Glaciation could thus possibly play an important role in regulating the highlands. This glacial erosion of course is strengthened during Quaternary glaciations (Fu et al., 2013; Zhang et al., 2016). We don't attribute the total removal of 2.5 km of rocks to pure glacial erosion, for which existing evidences show an initiation since only 2 Ma, it should be coupled with the other erosion processes, accompanying the regional uplift since 10 Ma.

The regional extent of glaciation during the Last Glacial Maximum (LGM) has been well documented in previous studies (Fu et al., 2013; Zhang et al., 2016), some of the key information from these studies has been adopted in our interpretation and will be illustrated with the new Fig. 7; we don't think it is necessary to document it on a DEM.

Line by line comments:

**Lines 18-20: I think that in the abstract you should suggest what could be the cause for rock uplift and Neogene exhumation for the BaimaXueshan massif.**

> This will be added as following: " We interpret exhumation of the massif as a response to regional uplift around the EHS due to the indentation of it since ~10 Ma." (line 17-18)

**Line 25. What depth (see also few comments below)?**

> The shallow segment steepens until a depth of 15 km and flattens to a depth of 17 km over a distance of 14 km.

**Line 36: How do you define low-relief surfaces? You should define what kind of relief threshold you use for a low-relief surface (at least to a first order) and document it by showing a proper DEM (please see also comment on Figure 1). This probably applies also to the abstract (line 20) where the low-relief surface is mentioned for the first time.**

> In this study, we follow the definition of low-relief surfaces as having a local relief of <600 m within a radius of 5 km and a local slope <10° according to Clark et al. (2006). We chose a Landsat image with expressive colors rather than a DEM, because it shows better the location and extent of low-relief surfaces across SE Tibet.

> This definition of low-relief surfaces will be added in the abstract as following: "These low-relief surfaces, with relief <600 m in a radius of 5 km, have all been previously interpreted as "relict surfaces"; i.e., remnants of a paleo-landscape that was formed at low elevation and subsequently uplifted (Clark et al., 2005; 2006)." (line 43-45)

> The threshold for low relief will also be specified in the abstract where it is first mentioned, as "Modelling results for the low-relief (relief <600 m), moderate-elevation (~4500 m) BaimaXueshan massif, east of the Mekong River, suggest regional rock uplift at a rate of 0.25 km/Myr since ~10 Ma, following slow exhumation at a rate of 0.01 km/Myr since at least 22 Ma.". (line 14-16)

**Line 51: "However, the direct link between river incision and regional tectonic uplift that is inherent in this interpretation has been disputed (Liu-Zeng et al., 2008; 2018; Nie et al., 2018)". I think you should comment this sentence here in introduction. What has been disputed and why? Please consider, that not all readers are familiar with the debate. I guess that this is fundamental for framing your research question.**

> This will be added as following: "However, the direct link between river incision and regional tectonic uplift has been challenged by others, who argued for a delay between crustal shortening, occurring mostly in the Eocene, and significant incision mostly during the Miocene in southeast Tibet (Liu-Zeng et al., 2008). A purely climatic forcing on river incision by intensified monsoonal precipitation at ~17 Ma was proposed by Nie et al. (2018)" (line 53-56)

**Line 61: In such regard, I think that you should already highlight in the introduction that Nie et al., interpreted the cooling pattern observed across the Mekong as an increase in precipitation rather than an indication of surface uplift (although that study is based on samples collected further south, so the correlation may not be straightforward)**

> This will be added as in the previous answer. (line 53-56)

Line 100: I think that you should highlight all these differences by showing a E-W oriented topographic swath profile across the Mekong and its flanks (BaimaXueshan and Kawagebo). This is fundamental to properly quantify river incision (which I guess has been discussed and quantified in several papers, but readers should not be forced to look for old literature unless strictly necessary). You can provide easily such a figure. I also think that you should present a N-S topographic swath profile (see also comment to line 36 and to figure 8).

> As answered to the first major comment, topographic profiles covering the BaimaXueshan and Kawagebo massifs (E-W oriented), as well as the Daocheng surface (NE-SW oriented) will be added to show the low-relief characteristic of both (see Figure 1d, e above). No N-S swath profile for BaimaXueshan will be necessary to demonstrate that. The quantification of river incision will be illustrated by a topographic profile across the Mekong River valley (Fig .7).
>
> In the revised version of the manuscript, we will add the quantification of incision made by previous studies based on thermochronological ages, as following: "In the upper and lower reaches of the Mekong River, AHe ages from the valley bottom cluster around 20-15 Ma and have been interpreted as recording >700 m of incision of the river in the interval ~17-14 Ma, linked to intensification of monsoonal precipitation (Nie et al., 2018)." (line 148-150)

Line 107: "In contrast, no clear estimation for the age of plateau uplift in the Three Rivers Region has been obtained" In contrast to what? In the previous sentences the uplift of the plateau is not discussed. I think that a summary about our knowledge of surface uplift/paleo-elevation history of the SE Tibetan Plateau should be reported somewhere. This kind of information is completely missing in the manuscript but is fundamental for setting up your modeling scenarios.

> To be clearer, we will modify line 107 as follow: "Compared to the timing of extrusion, no clear estimation for the timing of plateau uplift in the Three Rivers Region has been obtained." (line 120-121)
>
> We do not wish to expand this discussion to the uplift history of the SE Tibetan plateau, which is beyond the scope of this study. However, we do briefly discuss it in the previous section. (lines 57-61)

Line 108: Please show the location of the Jianchuan basin in a figure.

> This will be added in the Figure 1.

Line 181: Can you model a scenario consisting of a low-relief surface at low elevation, uplifted and incised at the same time (see for example Clark and Royden 2000)? I guess that should be something like your "incision" scenario but with surface uplift, where the low-relief areas are assumed to represent a landscape that has not yet adjusted to the new rock uplift rate. Or do you think that available thermochron. data do not support any longer this idea? I think that your study may contribute to this debate.

> Technically this is feasible, but the thermochronology data will not be able to distinguish this from our "incision" scenario, as they only constrain exhumation, not (surface) uplift. Moreover, the paleo-altimetry data along the Mekong River show that this part of plateau attained its present elevation in the late Eocene or Oligocene at the latest. (line 50-61)

Line 182: You assume that at 10 Ma there was already a plateau with an elevation like the modern one. Can you quickly report what data are supporting this scenario?

Several paleo-altimetry studies (Hoke et al., 2014; Li et al., 2015; Wu et al., 2018) show that the plateau in Three Rivers Region was already at its present elevation since at least the Oligocene. (line 50-61)

**Line 188: What does crustal-scale mean? Please clarify it. After reading the beginning of the following section I realized that the crustal scale fault scenario was used only for the Kawagebo Massif. I am wondering if you should anticipate this in the methods (where you talk about a generic fault) in order to provide a complete picture of the modeling scenarios (maybe the 6 lines of text at the beginning of the results section could go directly in the methods?).**

We add the qualification "rooting at a mid-crustal depth", and add the specification that this is implemented only for the Kawagebo massif, to this phrase to clarify this. (line 202-205)

In the method part, we just simply and explicitly explain the scenarios explored in this study and a brief explaining of the potential uses of Pecube, and in the following section we clarify the scenarios applied to each massif respectively, so we prefer to keep this part as present.

**Line 240: not clear to me how you quantified river incision (same applies to line 313).**

Please see answers to the third main comment.

**Line 345-350: Long sentence. Difficult to follow. Please rewrite it**

Agreed; it will be rewritten as following: "Surprisingly, both the previous 1D thermal-history modelling of Replumaz et al. (2020) and the 3D thermo-kinematic modelling reported here, imply that none of the tectonic events occurring in southeast Tibet before 7 Ma have been recorded as exhumation phases in the BaimaXueshan massif. Those events include extrusion of the Indochina block between ~34 and ~17 Ma (e.g. Leloup et al., 2001), significant erosion in the source region recorded by rapid sediment filling in Eocene basins downstream of the Three Rivers Region between 37-35 Ma (Gourbet et al., 2017), and shortening of these basins between 28 and 20 Ma (Cao et al., 2019). These results suggest that tectonic forcing in this massif has been negligible in driving its exhumation history before 7 Ma." (line 341-348)

**Line 360-370: I guess that this is a very important finding and tell us that the classic view (formation of low-relief surfaces at low elevation that will be uplifted and incised) decays. Should that be highlighted more? Unless the two surface are not the same object as I start suspecting (please see comment on Figure 8 and below).**

Indeed in this paper, we show that the BaimaXueshan massif in Three Rivers Region cannot be considered as a relict surface following the definition by Clark and Royden (2000), based on its significant recent exhumation (2.5 km since 10 Ma) recorded by relatively young thermochronology ages. We therefore conclude that not all low-relief surfaces can be explained by the mechanism invoked by Clark and Royden (2000), as also pointed previously by Zhang et al. (2016). But we can only emphasize different mechanisms for low-relief surface formation and we cannot generalize this mechanism to the generation of other low-relief surfaces across the whole of SE Tibet. We will rewrite the text to make this clearer.

**Lines 366-368: "Therefore, the BaimaXueshan massif cannot be considered to represent the same relict surface as the Daocheng granite, despite the fact that both have low relief (Fig. 1)". Following my previous comments: do they really have a similar topographic relief to be considered the same object, as written in the introduction? Maybe you should also present a NE-SW topographic swath profile that cover both regions and discuss it. The relief map shown in Figure 1c does not really suggests it. I can see**

that the BaimaXueshan is sandwiched between 2 main rivers, so its preservation potential is lower, but your Figure 1c seems to suggest that these two features look very different.

> On Figure 1c, the Daocheng surface does show less relief than some of the other surfaces. Nevertheless, the topographic profiles to be added (Fig. 1d, 1e) do confirm that the BaimaXueshan massif fits the criteria for a relict surface (local relief <600 m) set by Clark et al. (2006). We propose that other data should be involved to characterize relict surfaces, such as thermochronology ages in our study.

Line 371-379: "Such buzzsaw-like processes by glaciation could be active in the BaimaXueshan massif to smooth highlands, as numerous cirques, moraines and U- shaped valleys are observed across the massif, providing evidence for significant glacial erosion". The erosional mechanisms that can remove ca. 2.5 km of rocks and generate elevated low-relief surfaces are a crucial issue. Although this is not the main point of the manuscript the conclusions of this work have important geomorphologic implications. I think that the authors should provide a figure (probably just a DEM) documenting the glacial morphologies described in the text, or at least they should refer to previous publications documenting the processes (in Tibet or elsewhere) that may have generated such a topography. The other point that I think needs to be addressed is the timing of these erosional processes that should post-date (at least for 8 Millions of years) the onset of widespread high-altitude glaciations (ca. 2 Ma). Does it mean that first you formed a much higher topography and later you removed the material without any additional, significant cooling? In case, could you document it with yours 3D thermo-kinematic models?

> We conclude that moderate erosion accompanying the regional uplift due to northward indenting EHS from 10 Ma leads to this removal of 2.5 km overburden in the BaimaXueshan massif. We will rewrite this paragraph to read: "Glacial erosion and vigorous periglacial processes have been shown to be efficient in smoothing high-elevation regions above the glacial equilibrium line altitude (ELA) (Brozovic et al., 1997; Egholm et al., 2009; 2017; Hales and Roering, 2009). Zhang et al. (2016) have proposed that these processes could be active in southeastern Tibet and could lead to seemingly continuous low-relief high-elevation surfaces despite spatially differential and diachronous tectonic exhumation of 2-4 km between the surfaces. The mean ELA across southeast Tibet during the last maximum glaciation (LGM, ~20 ka) was estimated to ~4.6 km, much lower than the present-day ELA at ~5.4 km (Fu et al., 2013, Fig. 7). The ELA during previous glacial phases of the Quaternary would have been comparable to that of the LGM, with the average Quaternary ELA lying somewhere these values. Numerous cirques, moraines and U-shaped valleys are observed across the BaimaXueshan massif, providing evidence for significant glacial erosion (Fig. 8). Based on this observation as well as the maximum elevation of the massif (5200 m), which is close to close to the present-day ELA (5400 m), we suggest that buzzsaw-like (peri-) glacial processes could be active in the BaimaXueshan massif, such that any tectonic uplift bringing the elevation of the plateau above the ELA could trigger glacial processes that would smooth these highlands, as previously proposed by Zhang et al. (2016)." This now includes clear references to the processes we envisage, as well as a reference to a previous study envisaging this for SE Tibet.

Line 428: "which flattens at depth" What depth? Can you provide estimates? Could they correspond to any specific rock boundary?

> As answered to the comment on line 188 above, the shallow segment steepens until a depth of 15 km and flattens until 17 km over a distance of 14 km, as shown in table 2, line 305 and on Figure 5c. But unfortunately, given the lack of geophysical data in the study region, we cannot tell whether this depth corresponds to any identified rock boundary.

Figure 1B: What is the benefit of showing a Landsat image rather than a DEM? Frankly speaking I do not find such a Landsat image useful, especially, if you are not familiar with the region. The scale of figure 1a does not allow appreciating the topographic characteristics and the position of the paleosurfaces. I guess that you need to use a DEM as background of figure 1A. Especially after reading the conclusion for the BaimaXueshan massif. I also think that you need some topographic swath profiles (see comments above).

> The Landsat image, in our view, better shows the geomorphology of SE Tibet, with cyan marking the high snow-covered areas mainly around the EHS, green marking incised and vegetated river valleys, and brown/purple/yellow marking the low-relief surfaces, which are additionally outlined by black lines. Swath profiles will be added, as suggested by reviewer1, to better show the geomorphological characteristics of the relict surfaces, particularly the BaimaXueshan and Daocheng surfaces.

Figure 2: I am sorry, but my eyes cannot really differentiate the Triassic Qiangtang from the Yidin. Could you please increase the color difference? Also note that some black lines are thicker, apparently without any specific reason, than others. Finally, Eocene is not properly spelled; please correct it. What about showing a simplified cross section? Are there available in literature? They could help following the geological setting.

> This will be corrected, such precise identification of different Triassic units is not needed in our study. This region is highly deformed due to N-S thrust shortening and subsequent strong N-S shearing during extrusion, no studies have attempted to reconstruct a cross-section here yet (although we are working on one). On Figure 8, we show a crude attempt to reconstruct the crustal geometry of the fault.

Table 2: Does the sketch has a vertical scale that I respected? What is the level of incision with respect to the initial plateau? This should be discussed somewhere (see comment to line 240) to better appreciate the contribution of the Mekong on the observed cooling history.

> Those sketches are used to show the concept of different scenarios and are not precisely scaled, the real scale is shown clearly in the 3D visualization of Figure 3c and 5c.

> A schematic topographic profile across Kawagebo and BaimaXueshan massifs will be added to show the quantification of river incision as compared to the total exhumation (Fig. 7).

Figure 5: What are the modeling shortening rates over the last 1.5 Ma for the Kawagebo massif? Are these rates compatible with the lateral shearing rates recorded by GPS?

> The modeled shortening rate since 1.5 Ma is about 1.86 mm/yr. As the Kawagebo is located in a constraining bend between dextral Parlung and Zhongdian faults, it absorbs stronger shortening than regionally and also shows high vertical motion velocity (1.7 mm/yr). Large horizontal velocity should be measured too along those strike-slip faults, however, no lateral shearing rates based on local GPS velocity have been determined to show spatial variations of horizontal and vertical velocity.

Figure 8: From this figure the low-relief surface of the BaimaXueshan massif does not look like a typical uplifted low-relief landscape (please see also my comments above). I think that a clear definition of low-relief surface is fundamental. To me the landscape of the BaimaXueshan massif and the Daocheng area looks totally different and the large difference in cooling ages observed across these 2 surfaces is not surprising. I also suppose that the erosional mechanism that led to the development of these "surfaces" is very different (glacial vs fluvial?) and operated on different time scales. I start thinking that you are comparing surfaces that should not be compared. Such a comparison is misleading because these two

surfaces are not the same landform. I understand that this is one of your major conclusions; my concern is that your starting point (i.e., these 2 surfaces are the same thing) does not appear to be supported by geomorphic evidence, unless you will document with a couple of figures.

See answers to the 1st major comment and line 36, 366-368 comments.

---

## Author Comment (AC2) · 21 Dec 2020

We thank reviewer Massimiliano Zattin for his constructive and helpful comments. Below, we respond to the reviewer's line-by-line comments (response indented, line numbers in red mark the modifications in revised manuscript).

Line by line comments:

Lines 52-54. The age range for this rapid exhumation event is very large (from 20 and 60 Ma). I do not think that it is possible to talk about a single "phase" as more than one could have been taken place in a single region.

> The phrase will be changed to "Other thermochronologic studies have also provided evidence for earlier phases of rapid exhumation, the timing of which varies regionally between 30 – 20 Ma in the Longmenshan (Wang et al., 2012; Tan et al., 2014), to 40 – 30 Ma in the Yalong thrust belt (Zhang et al., 2016) and ~60 – ~40 Ma in the BaimaXueshan massif (Liu-Zeng et al., 2018). The latter has been linked to uplift of the Southeast Tibetan plateau, consistent with paleo-elevation data implying that the plateau has been close to its present-day elevation since the Late Eocene - Oligocene (Hoke et al., 2014; Li et al., 2015; Wu et al., 2018)." (line 56-60)

Lines 75-78. As it is written here, it is not clear if the role of tectonics in the exhumation of Kawagebo is derived from literature or is one of the output of this work. I would better specify which are the goals of the paper.

> Replumaz et al. 2020 have shown that exhumation was so rapid in the Kawagebo that it implies tectonic forcing. They speculate a local fault underlying the Kawagebo massif, which has not been observed in the field. In our study, by using the same data, we explore this hypothesis and could constrain the geometry of this fault. It is one of the specific goals of this paper to determine the geometry of the fault along the Mekong, but also to examine the influence of Mekong incision on exhumation, by doing more complex modeling of published data using Pecube.

> This will be clarified by adding: "The Kawagebo massif is thought to have been rapidly exhumed recently (<10 Ma) due to motion on a local thrust fault in a restraining bend between two regional-scale strike-slip faults (Replumaz et al., 2020, Fig. 1b, 1c)" (line 80-82)

Figure 1. Separation of AFT and AHe ages in two separate maps is good in terms of readability but it forces to move from one figure to the other to have a complete picture of exhumation ages. Is it not possible to merge all the ages in a single map?

> We separate AFT from AHe ages because they do not convey the same information: the AHe map shows more clearly the Miocene rapid exhumation phase where AHe ages are <15 Ma, while the AFT map shows very clearly the relict surfaces where AFT ages are >50 Ma. Furthermore, a map with both AFT and AHe ages is shown in the new figure 9 (previous figure 7).

Line 85 (caption of figure 1). Why grey outlines? I see only black lines around these surfaces.

> It was grey lines on Figure 1b to show the same relict surfaces as in Figure 1c. But we made them black as in Figure 1c and will modify the legend as: "Black outlines delimit low-relief relict surfaces with relief <600 m, as mapped by Clark et al. (2006)." (line 93)

**Line 103. The name "AilaoShan fault" is not in the map. Moreover, the "Red River" marks a fault and not a river.**

The name "Ailaoshan fault" will be replaced by "Ailaoshan-Red River shear zone" and it will be added in Figure 1a, together with "Red River". To be clearer, we will modify as following: "From the late Eocene to the early Miocene, the dominant driver of deformation in SE Tibet was the extrusion of the Indochina block along the left-lateral AilaoShan-Red River shear zone following the Red River (Fig. 1a), subsequently inverted along the right-lateral Red River fault since ~5-10 Ma (Leloup et al., 1995, 2001; Replumaz et al., 2001; Fyhn and Phach, 2015) (Fig. 1b)." (line 114-117)

**Line 107. This sentence is not related to the previous one as they deal with very different topics. So, why "in contrast"?**

The timing of the uplift of the plateau is not well constrained when comparing to the timing of the extrusion.

To be clearer, we will modify line 107 as follow: "Compared to the timing of extrusion, no clear estimation for the timing of plateau uplift in the Three Rivers Region has been obtained." (line 119-120)

**Lines 108-111. This sentence is not well connected to the previous ones. In general, this paragraph appears as a collage of sentences with no clear relationships between them.**

This paragraph presents the deformation history since Late-Eocene, focusing on the most important studies who show significant uplift phases in different zones across or near the Three Rivers Region. In the discussion part, we compare them with our modelling results, showing that none of these phases has been recorded as exhumation phases by the thermochronology dataset.

To be clearer, this paragraph will now read: "Rapid sediment filling of the Jianchuan basin (Fig. 1b), located downstream of the Three Rivers Region, around 37-35 Ma demonstrates significant erosion and suggests uplift in the source region just predating extrusion (Gourbet et al., 2017). The Jianchuan basin subsequently experienced significant exhumation along thrust faults between ~28 and 20 Ma (Cao et al., 2019), suggesting regional uplift at that time." (line 120-123)

**Line 112. How can a shear zone join a river? Furthermore, the AilaoShan-Red River shear zone is not marked in the map of figure 2a.**

The Ailaoshan-Red River shear zone is prolongated along a Jurassic red-wine colored clastic formation, which is intensively sheared and roughly follows the Mekong river, as shown and discussed in Replumaz et al. (2020).

We will re-write the sentence as following: "the AilaoShan-Red River shear zone is prolongated along a distinctive and intensively sheared Jurassic red-wine coloured clastic formation, following the Mekong River". (line 124-125)

We will mark the ASRRSZ on Figure 1a, 2a. (as shown in the answer to the comment of Figure 2a below)

**Lines 116-118. The only Eocene deposits visible in map of fig. 2A are located east of the city of Deqing. Is it just a matter of scale? In the text you describe "several thrusts affecting Eocene basins".**

There are, indeed, only 3 Eocene basins on the map of Figure 2a. Other Eocene basins located further to the north or south of the Three Rivers Region could not be shown; the most emblematic one is the Jianchuan basin (Fig. 1), located south of the map of Figure 2, and also affected by thrusts (Cao et al 2019).

**Figure 2A. There is a thin red line in the top of the map, nearly parallel to the Yangtze river, that is probably not correctly drawn. In the legend, check the word "Eocene". The colors of Triassic Yidun and Qiangtang formations are very similar**

All those will be corrected in figure 2A.

**Line 153. ". . .of the onset of this rapid exhumation phase". Are you referring to the 8 Ma or 1.5 Ma step?**

We are referring to the exhumation since 8 Ma here. The sentence will be changed to "Quantitative time-temperature inversion of the Kawagebo ages suggests rapid exhumation since at least 8 Ma, with no clear estimate of the onset timing, and followed by an acceleration after ~1.5 Ma (Replumaz et al., 2020)." (line 162-164)

**Lines 155-156. Which structure? The stepover? Actually it is defined by faults and I see that these faults have been mapped and here described.**

This structure refers to the local thrust fault in a restraining stepover between two dextral faults. The sentence will be changed to "However, this local thrust fault, inferred to be related to reactivation of regional north-south trending thrusts, has not been documented in the field." (line 166-167)

**Line 158. What do you mean with "collision period"? Tectonics here was changing through time so, for example, the Eocene is marked by extensional basins.**

Collision, in the context of the Tibetan Plateau, refers to the Indian-Eurasian continent collision, which is ongoing since ~55 Ma and continuously set the fundamental tectonic regime for SE Tibet, even though local-scale extension is recorded during the Eocene.

The sentence will be changed to "Therefore, additional work is needed in this region to resolve the exhumation history of the low-relief mean-elevation BaimaXueshan and the high-relief high-elevation Kawagebo massifs during the India-Asia collision period, in order to distinguish the effects of regional plateau uplift, incision of the Mekong River, and uplift along local tectonic structures." (line 167-169)

**Lines 186-187. The second part of the sentence is not very clear. Can you better explain the meaning of "transition times"?**

Transition time is a term used in Pecube modeling to represent the timing between two exhumation phases.

A phrase will be added to explain this concept: "Different transition times mark the timing of exhumation or topographic changes between any two different phases." (line 189-190)

**Table 1. It is not clear if the references are related to the left or the right parameters. Or both?**

In this table, parameters without references are default values from Pecube, but to avoid confusion, we will add Braun et al. (2012) and Chen et al. (2014) as reference for them.

**Lines 222-223. This conclusion is referred to the steady-state scenarios only? Or is it more general?**

Yes, this only refers to those steady-state scenarios with three thermochronometers. The models with three thermochronometers and different exhumation phases are designed to decipher the exhumation history since 110 Ma, trying to reveal the important exhumation phases regionally observed during India-Asia collision. But, surprisingly, none of these regional tectonic events has been recorded in our modelled exhumation history. That's why we conclude that BaimaXueshan only experienced a single rapid exhumation phase since ~7 Ma.

Lines 230-232. I agree on the focus on AHe and AFT but... what happens if you exclude the ZHe data also in the previous scenarios? Such a change in the input data could have relevant effects on the outputs?

> The ZHe ages in BaimaXueshan are all >75 Ma, showing that those samples must have been exhumed at a very slow rate before the recent rapid exhumation phase (<7 Ma). They therefore do not influence much the young exhumation phase deduced from AFT and AHe ages.
>
> We will modify the sentence to clarify that: "We finally tested a plateau scenario, including both regional rock uplift and incision, fitting only the AHe and AFT ages younger than 22 Ma to concentrate on the Neogene history, considering that the steady-state models, including the old ZHe ages, adequately predict the early history." (line 236-238)

Lines 289-290. Why since 10 Ma?

> We choose to start our modelling at 10 Ma, because the maximum ZHe age, considering the error bar, is ~9 Ma, and allowing a predicted age slightly older than it, as observed for the Baimaxueshan (Fig. 3b).

Line 299. Given the dipping angle, I would not use the term "thrust" for this fault.

> Indeed, in this tectonic scenario with a simple planar fault (one-segment), the dipping angle is 85°, we will change it to reverse fault in this paragraph. (line 287-290)

Figure 5. I am a bit confused... Each arrow yields a color which should be related to an exhumation rate... but is this figure associated to a specific time? In fact, here you write about the presence of different exhumation phases.

> In Pecube, the velocity field over a fault is constant during one phase of exhumation, so Figure 5 is valid between 1.5 and 0 Ma. This will be clarified in the legend of Figure 5 as following: "The arrows show velocity field generated by movement along the fault, during the rapid exhumation phase, since 1.5 Ma;" (line 318-319)

Lines 345-350. I am not sure that this sentence is correct as your model is starting at 22 Ma. All the events cited here are occurring mostly before 22 Ma.

> This part discusses the steady-state scenario including three thermochronometers since 110 Ma, where deformation phases before 7 Ma could not be revealed in exhumation history of BaimaXueshan. The modelling of Neogene history since 22 Ma is discussed in the following paragraph.
>
> Modification will be made to clarify it in the beginning of this paragraph: "For the BaimaXueshan massif, east of the Mekong River, our best-fit model, exploring the exhumation history since 110 Ma, shows rapid regional exhumation at a rate of 0.42 km/Myr since ~7 Ma, succeeding a phase of slow regional exhumation at a rate of 0.04 km/Myr before that." (line 331-333)

Line 352. So the paragraph above is related to the models starting before 22 Ma? This is not very clear. But, if this is true, why are you discussing the models starting before 22 Ma if you write that these are not resolving well the dataset?

> One of our first conclusion is that the uplift phases before 7 Ma during the India-Asia collision is not expressed by the exhumation history of the BaimaXueshan massif. The age reconstruction shows that this model resolves relatively well the overall dataset covering a 110 Ma history. The Neogene history with young AHe and AFT ages since 22 Ma is dedicated to explore the recent exhumation and Mekong river incision.

**Line 368. This is not the definition of "relict surface" or, better, its definition should not be related to time of collision (Clark et al. do not give any definition like that).**

Clark et al. (2005) defined relict surfaces as "remnants of a paleo-landscape that was formed at low elevation and subsequently uplifted" implying low exhumation rate during the collision. Our compilation of thermochronological ages and our modelling results lead us to add the age constraint (AFT >50 Ma) to define a low-relief surface as a relict surface. A surface with younger AFT ages should have experienced significant exhumation during the collision, such as the Baimaxueshan.

We will add a precision to the text: "We conclude that a relict surface *sensu-stricto*, i.e., remnant of a paleo-landscape that was barely affected by exhumation during India-Asia collision (Clark et al., 2005), should show AHe ages older than 50 Ma, and that not all low-relief surfaces in southeast Tibet can be classified as a relict surface." (line 361-364)

**Lines 385-386. Actually there are not so many data between the Parlung fault and the Longmucuo-Shuanghu suture and young ages are widespread also more to the south and to the north.**

Many existing studies focus on the Eastern Himalayan Syntaxis or neighbouring regions, but ages are more or less well spread over the entire region. To first order, our correlation between an elevation contour of 5000 m and the rapid uplift phase since at least 10 Ma in and around the EHS, marked by AHe ages <15 Ma and AFT ages <50 Ma, is valid. Most young ages to the northeast of the suture are either due to supplementary local tectonic exhumation, such as close to the Litang fault, or strong river incision, with AHe ages <15 Ma along the Yangtze River.

**Line 388. Define the acronymn EHS.**

EHS stands for Eastern Himalayan Syntaxis, as defined in line 11.

**Lines 390-391. An extrapolation of the present-day velocity field to 10 Ma ago can be chancy. On the other hand, if your results are coherent with the present-day velocity field, you can infer that plate kinematics has not changed since then.**

There is no extrapolation of the present-day velocity field, which is measured on very different time scale from our modelling. We just use the GPS data to show the present velocity field related to the ongoing indentation of India on a much wider scale across Tibet than our modelling.

**Line 395. Not sure about the use of "far-field" as this area is along the border of EHS.**

We remove the word far-field, and modify the sentence as: "Moderate uplift along those thrusts could be due to the distance from the EHS, with a west-to-east gradient of decreasing exhumation and erosion rates across the Three Rivers Region (Yang et al., 2016)." (line 395-397)

**Line 413. What do you mean with "since 10 Ma at river level"?**

Both the amount of tectonically controlled exhumation and the amount of river incision are spatially variable; the former is controlled by the modelled geometry of the fault and the latter depends on the modern elevation. In order to clarify this, the phrase will be changed as follows: "The average topographic lowering due to river incision at the elevation of the samples is ~2 km, accounting for ~25% of the total exhumation of the massif (Fig. 7)." (line 414-415)

**Lines 424-426. Your model is clearly designed to verify the amount of tectonic exhumation along a thrust and the results to confirm that this model is working well. Furthermore, no acceleration of exhumation since 1.6 Ma is occurring in the BaimaXueshan massif. As a whole, these data seem to**

indicate that glacial erosion, if present, was of minor importance. This is an interesting outcome that could be emphasized.

> It's true that the recent acceleration of exhumation since 1.6 Ma of Kawagebo massif is not modelled in BaimaXueshan massif. But as discussed in line 369-379, we could not exclude the potential role of glaciation in the exhumation of BaimaXueshan, as numerous cirques, moraines and U-shaped valleys are observed across the massif. And until now, unfortunately, we could not quantify the contribution of glacial erosion on exhumation due to the limit of Pecube of not being able to model the glaciation processes.

**Line 442. What do you mean with "postulated crustal geometries of fault"? Are you referring to the "black" faults or the active ones?**

> This refers to the active faults in red and will be clarified as follows: "Also shown are the crustal geometries of faults (in red) active in the last 10 Myr." (line 438)

**Lines 454-456. Once again I would stress the fact that data and your modelling focus on the "young" (i.e. Neogene) part of the history. So I am not very sure that you can affirm that tectonic events were negligible before 10 Ma**

> The steady-state scenario of three thermochronometers since 110 Ma only resolves a rapid exhumation phase since 7 Ma, showing that the regional deformation phases, such as large scale motion on shear zones (34-17 Ma) and Eocene basin filling (37-35 Ma), have not been expressed in the exhumation history of BaimaXueshan.

> This is our first conclusion in the discussion part, some modification will be made to clarify it: "For the BaimaXueshan massif, east of the Mekong River, our best-fit model, exploring the exhumation history since 110 Ma, shows rapid regional exhumation at a rate of 0.42 km/Myr since ~7 Ma, succeeding a phase of slow regional exhumation at a rate of 0.04 km/Myr before 7 Ma." (line 331-333)

**Line 456. Can you put a number? Otherwise the meaning of "moderate" is ambiguous.**

> Yes, 0.25 km/My of exhumation will be added. (line 463)

**Line 460. I would emphasize also the inferred acceleration at 1.5 Ma and the exhumation rates that jump to values higher than 1 km/Ma (one order of magnitude).**

> Yes, this will be added as following: "Total tectonically driven exhumation since ~10 Ma at a rate of 0.45 km/my, with an acceleration to 1.86 km/My since 1.6 Ma, is estimated to be 6.1 km, with incision of the Mekong River contributing only ~25% of the total exhumation." (line 467-469)